# Cyclin Kinase-independent role of p21[CDKN1A] in the promotion of nascent DNA elongation in unstressed cells

**Sabrina F Mansilla[1], Agustina P Bertolin[1], Valérie Bergoglio[2,3,4,5†], Marie-Jeanne Pillaire[2,3,4,5†], Marina A González Besteiro[1], Carlos Luzzani[6], Santiago G Miriuka[6], Christophe Cazaux[2,3,4,5‡], Jean-Sébastien Hoffmann[2,3,4,5], Vanesa Gottifredi[1]\***

[1]Fundación Instituto Leloir-Instituto de Investigaciones Bioquímicas de Buenos Aires, Consejo de Investigaciones Científicas y Técnicas, Buenos Aires, Argentina; [2]Centre de Recherches en Cancérologie de Toulouse, Toulouse, France; [3]INSERM, Universite Paul Sabatier-CNRS, Université de Toulouse, Toulouse, France; [4]Laboratoire d'Excellence TOUCAN, Toulouse, France; [5]Equipe labellisée La Ligue contre le Cancer, Toulouse, France; [6]Laboratorio de Investigaciones Aplicadas en Neurociencias, Fundación para la Lucha contra las Enfermedades Neurológicas de la Infancia, Belén de Escobar, Argentina

**Abstract** The levels of the cyclin-dependent kinase (CDK) inhibitor p21 are low in S phase and insufficient to inhibit CDKs. We show here that endogenous p21, instead of being residual, it is functional and necessary to preserve the genomic stability of unstressed cells. p21depletion slows down nascent DNA elongation, triggers permanent replication defects and promotes the instability of hard-to-replicate genomic regions, namely common fragile sites (CFS). The p21's PCNA interacting region (PIR), and not its CDK binding domain, is needed to prevent the replication defects and the genomic instability caused by p21 depletion. The alternative polymerase kappa is accountable for such defects as they were not observed after simultaneous depletion of both p21 and polymerase kappa. Hence, in CDK-independent manner, endogenous p21 prevents a type of genomic instability which is not triggered by endogenous DNA lesions but by a dysregulation in the DNA polymerase choice during genomic DNA synthesis.

**\*For correspondence:**
vgottifredi@leloir.org.ar

†These authors contributed equally to this work

‡Deceased

**Competing interests:** The authors declare that no competing interests exist.

## Introduction

The p21 protein (also known as p21[CDKN1A] and p21[Cip1/Waf1]), is a member of the family of cyclin-dependent kinase (CDK) inhibitors (CKIs) which has long been known for its ability to consolidate the G1 and G2 arrest after DNA damage caused by genotoxic agents, such as γ irradiation (γ IR) (*Brugarolas et al., 1995*; *Bunz et al., 1998*; *Deng et al., 1995*; *Dulic et al., 1998*). As p21 has no enzymatic domain, it was not surprising to discover that a robust increase in p21 protein levels is required to achieve efficient CDK inhibition in response to DNA damage (*Boulaireet al., 2000*; *Cai and Dynlacht, 1998*). Such observations have led to the widely-held assumption that the lower amounts of p21 in cycling cells are residual and insufficient to achieve any biological relevant function (*Bertolin et al., 2015*; *Soria and Gottifredi, 2010*).

However, p21 levels in cycling cells are not null. Albeit p21 does not efficiently inhibit CDK activity in cycling cells (*Cai and Dynlacht, 1998*) it could still regulate CDK-independent processes. CDK-independent functions of p21 could rely on its proliferating cell nuclear antigen (PCNA)-interacting

**eLife digest** Cancer develops when cells in the body mutate in ways that allow them to rapidly grow and divide. To protect cells from becoming cancerous, various molecules act like guardians to prevent cells from dividing when their DNA is damaged, or if they are short of energy. Other guardian molecules monitor the DNA copying process to ensure that the newly-made DNA is as identical as possible to the original DNA template.

A protein called p21 belongs to the first group of guardian molecules: DNA damage triggers the production of p21, which prevents the cell from copying its DNA. This role relies on a section of the protein called the CDK binding domain. Cells that have already started to copy their genetic material also have low levels of p21.

Mansilla et al. used human cells to investigate whether p21 is also involved in the process of copying DNA. The experiments show that the low levels of p21 act to increase the speed at which the DNA is copied. This activity helps to ensure that all of the cell's DNA is copied within the time available, including sections of DNA that are harder to copy because they are more fragile and prone to damage. This newly identified role does not involve the CDK binding domain, but instead requires a different section of the p21 protein known as the PCNA interacting region.

Mansilla et al. propose that p21 plays a dual role in protecting us from developing cancer. The PCNA interacting region is also found in other proteins that are involved in copying DNA. Therefore, a future challenge is to find out how these proteins interact with each other to ensure that cells accurately copy their DNA in a timely fashion.

region (PIR) located on the C-terminus of p21, which binds the interdomain connecting loop (IDCL) of PCNA with high affinity ([*Prives and Gottifredi, 2008*] and references there in).However, no role for the p21/PCNA complex formation has been yet described. On the contrary, research has focused only on the biological relevance of disrupting the p21/PCNA interaction.

As DNA polymerases (pols) also bind the IDCL of PCNA through PIR or PIP (PCNA interacting protein) boxes, p21 should be capable of negatively regulating all PCNA-dependent DNA synthesis processes in cells (*Moldovan et al., 2007*; *Tsanov et al., 2014*). In fact, in vitro experiments demonstrated that the large excess of p21´s PIR inhibits the interaction of PCNA with DNA replication and nucleotide excision repair (NER) factors (for original papers refer to [*Prives and Gottifredi, 2008*]), impairing replication- and repair-associated DNA synthesis respectively (see examples in [*Cooper et al., 1999*; *Gottifredi et al., 2004*; *Prives and Gottifredi, 2008*]). However, the amount of p21 used in such experiments were much higher than the maximal p21 levels that can be accumulated in cells, even after genotoxic treatments (discussed in [*Prives and Gottifredi, 2008*]).The overexpression of p21 to levels in the range of those induced by genotoxins, inhibits neither replication- nor repair- associated DNA synthesis events (*Soria et al., 2008*) which are mostly dependent on replicative DNA pols (*Burgers, 1998*; *Soria and Gottifredi, 2010*). These data all together indicate that in cycling cells, physiological levels of p21 are not capable of inhibiting PCNA-dependent DNA synthesis by replicative DNA pols, even when p21 is induced by external stress.

However, PCNA-dependent synthesis by other DNA pols could be inhibited by endogenous p21. In fact, endogenous p21 levels drop dramatically after ultraviolet irradiation (UV) and Methyl methane sulfonate (MMS) treatments (*Soria et al., 2006*). We have previously shown that p21 downregulation after UV facilitates nascent DNA elongation across UV-damaged DNA templates by enabling the recruitment of alternative (alt) DNA pols to replication factories (*Mansilla et al., 2013*). Strikingly, UV irradiation couples translesion DNA synthesis (TLS) by alt DNA pols with the activation of the CRL4$^{Cdt2}$ E3 ligase at replisomes (*Havens and Walter, 2011*; *Nishitani et al., 2008*; *Soria and Gottifredi, 2010*). The CRL4$^{Cdt2}$ E3 ligase binds and degrades p21 *only* when it is complexed with chromatin-associated PCNA (*Abbas et al., 2008*; *Havens and Walter, 2011*). The list of genotoxic treatments that triggers p21 proteolysis has expanded lately and includes UV, MMS, cisplatin, hypoxia, hypoxia mimicking factors, hydroxyurea (HU), aphidicolin (APH), hydrogen peroxide, and potassium bromide (*Savio et al., 2009*). In conclusion, the degradation of endogenous p21 at replication sites in S phase allows full TLS activation or fork-restart when required.

While the above mentioned reports demonstrate the relevance of disrupting p21-PCNA interaction in cells, no report has ever addressed the relevance of the PCNA-p21 complex in cells. Here we report that endogenous p21 localizes at replication factories through PCNA binding, thereby avoiding DNA polymerase κ (Pol κ) to be recruited at replication factories. Surprisingly, in contradiction with its function as a negative regulator of CDKs, p21 facilitates S phase progression; that is p21 promotes nascent DNA elongation.The DNA replication defects caused by p21-depletion caused accumulation of replication stress markers, such as γH2AX and 53BP1, instability of common fragile sites and micronuclei (MN) accumulation. Interestingly, all the replication defects observed in p21-depleted cells were eliminated when Pol κ was depleted, and were also complemented by a p21 mutant with an intact PCNA binding domain and a disrupted CDK binding site. Collectively, our data demonstrate that, although expressed at low levels in S phase, p21 fine-tunes the dynamics of DNA replication by regulating Pol κ loading to replisomes. Therefore, while the CDKs/p21 interaction is crucial to the cellular response to DNA damage, the PCNA/p21 interaction prevents the accumulation of DNA-damage independent genomic instability in unstressed cycling cells.

## Results

### p21 localizes to replication factories in cycling cells

The limited amounts of p21 in cycling cells allow CDK-dependent cell cycle progression (*Kreis et al., 2014*). Indeed, p21 levels in cycling cells are not null and can be detected on EdU positive cells with p21 specific antibodies (*Figure 1A and B*) as reported recently (*Coleman et al., 2015*). Remarkably, while p21 levels are at the lowest in S phase (*Figure 1—figure supplement 1A,B*), they are sufficient to impair TLS-dependent DNA synthesis (*Mansilla et al., 2013*; *Soria and Gottifredi, 2010*) if not degraded after UV irradiation (*Figure 1—figure supplement 1A, B*). Notably, the function of p21 during unperturbed cell cycle progression remained unknown. A hint of such function was revealed by a Proximity ligation assay (PLA) which revealed a chromatin bound PCNA/p21 interaction in cycling cells. Such complexes resisted a mild extraction with CSK buffer which removes proteins unbound to chromatin (*Figure 1C–D*). Consistent with our previous findings, the percentage of cells with PLA spots was reduced by UV irradiation and PLA spots were not detected upon p21 depletion (*Figure 1C–D*). In agreement, endogenous p21 colocalized with PCNA (*Figure 1E and F*) and EdU-labelled replication factories (*Figure 1—figure supplement 1C*). The colocalization of p21 and GFP-PCNA became more evident following removal of proteins unbound to chromatin (*Figure 1—figure supplement 2A*). We next evaluated the requirement of the p21 PIR region for p21-PCNA colocalization. To this end, we transfected cells with either p21 or p21^PIPMut*, bearing an intact or a disrupted PIR, respectively (*Mansilla et al., 2013*; *Soria et al., 2008*). The disruption of the CDK-binding site by point mutations in both constructs (*Mansilla et al., 2013*; *Soria et al., 2006*, *2008*), prevented the arrest outside S phase expected after p21 overexpression (*Figure 1—figure supplement 2B,C*). Similar to endogenous p21, overexpressed p21 localized to replication factories (*Figure 1G* and *Figure 1—figure supplement 3A,B*). However, p21^PIPMut* lost its ability to form foci at replication factories (*Figure 1H* and *Figure 1—figure supplement 3C*), did not colocalize with PCNA (*Figure 1—figure supplement 3C*) and showed reduced chromatin retention (*Figure 1—figure supplement 3D and E*). Hence, the PIR of p21 is required for the localization of p21 to replication factories in cycling cells.

### Endogenous p21 preserves DNA replication homeostasis in cycling cells

On the basis of the localization of p21 to replication factories, we speculated that p21 could regulate the DNA replication dynamics during S phase. To test this hypothesis, we used p21-depleted U2OS osteosarcoma cells (*Figure 2A*). The number of cells positive for EdU and with PCNA bound to chromatin increased after p21 depletion (*Figure 2B and C*). An enrichment in PCNA focal distribution corresponding to mid-to-late S phase (*Essers et al., 2005*; *Rey et al., 2009*), was found in p21-depleted samples (*Figure 2D*) which suggested a defect intrinsic to S phase and independent from the G1/S transition.

Therefore, we evaluated DNA replication parameters specific to S phase in p21-depleted U2OS cells by means of the DNA spreading technology. Nascent DNA was labelled with a 10-min pulse of CldU (Chlorodeoxyuridine), followed by a 30-min pulse of IdU (Iododeoxyuridine). After

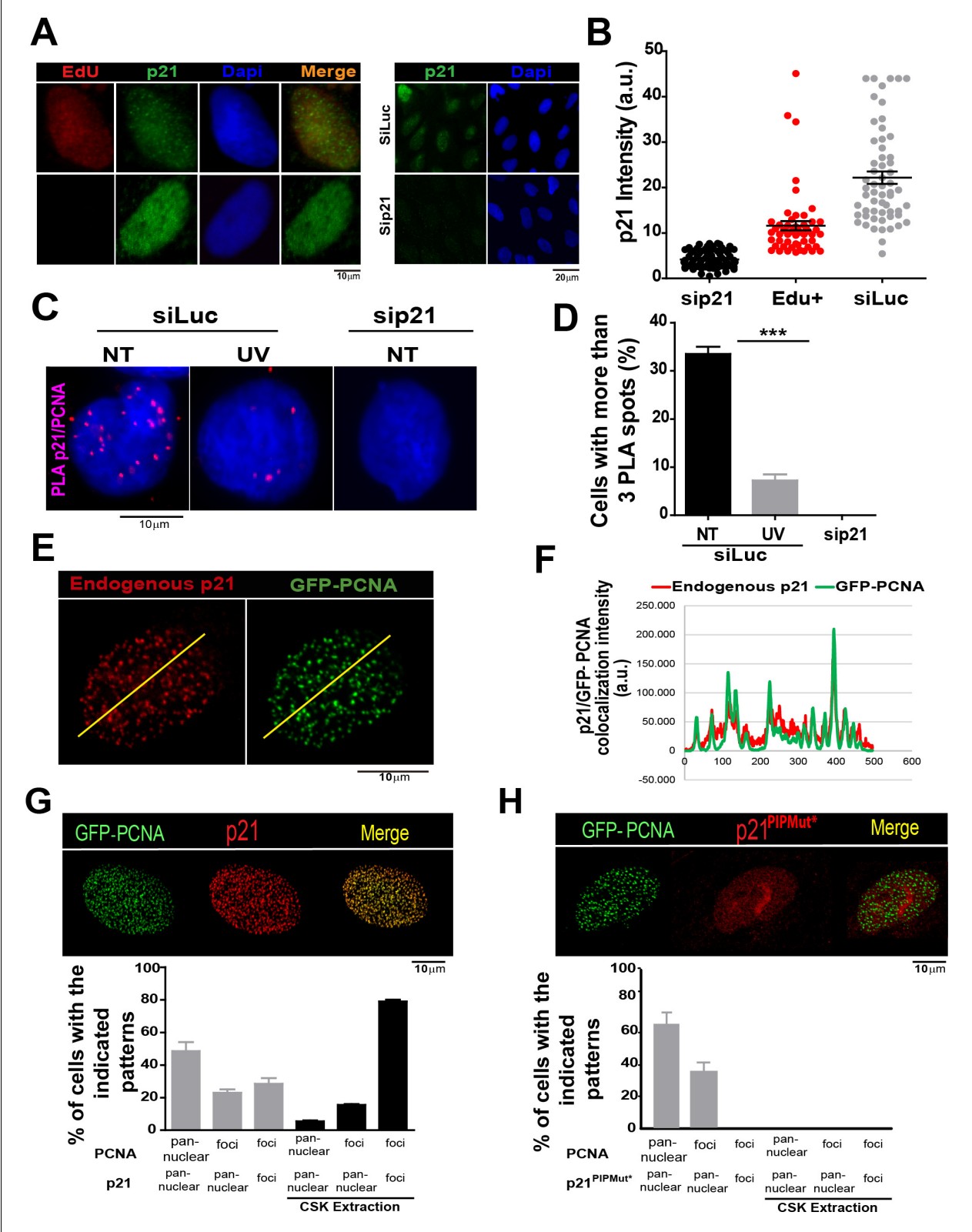

**Figure 1.** The PCNA interacting region of p21 facilitates the recruitment of p21 to replication factories in cycling cells. (**A**) Representative images of p21 in EdU positive and negative cells (left panel) and from U2OS cells transfected with control siRNA (siLuc) or sip21 (right panel). (**B**) p21 intensity in the indicated samples. Nuclei were counterstained with DAPI. 70 nuclei/sample; two independent experiments were performed. (**C**) Representative images from Proximity ligation assay (PLA) performed after mild extraction on the indicated samples. (**D**) Quantification of PLA experiments described in **C**. 100
*Figure 1 continued on next page*

*Figure 1 continued*

nuclei/sample; two independent experiments were performed. (E) Colocalization of p21 foci with GFP-PCNA which reveal replication factories (mild extraction was applied). (F) Profiles of signal intensity along an arbitrary line (showed in E) drawn across the nuclei. (G and H) Samples were transfected with the indicated p21 mutants. Representative images are shown. Samples treated or not with CSK buffer were classified into the indicated categories.100 nuclei/ sample; two independent experiments were analysed. For all figures in this manuscript: significance of the differences are: *p<0.1; **p<0.01; ***p<0.001. When the difference is not statistically significant, the p value is not shown. Error bars represent SEM (standard error of the mean).

The following figure supplements are available for figure 1:

**Figure supplement 1.** Endogenous p21 localizes to replication factories.

**Figure supplement 2.** Chromatin bound p21 is localized at replication factories.

**Figure supplement 3.** The PIR domain is required for p21 recruitment to replication factoriesU2OS cells were transfected with GFP-PCNA and p21 or p21[PIPMut*] respectively.

denaturalization, stretching and labelling with specific antibodies, DNA track lengths were quantified. Surprisingly, p21 loss resulted in a reduced track length suggesting that p21 is required for a proper elongation of the replication forks (representative tracks and fields are shown in *Figure 2E–F*). These findings were confirmed in an additional cellular model, the HCT116 p21 null cells which were compared with the HCT116 p21+/+ counterparts (*Figure 2—figure supplement 1A–C*). Such findings were unexpected from the perspective of the prevalent notion of p21 as a negative regulator of the cell cycle. Since defective fork elongation is generally coupled with increased origin firing (*Blow et al., 2011*; *Pillaire et al., 2007*; *Techer et al., 2016*), we measured the origin frequency as we did in the past (*Vallerga et al., 2015*) [number of red-green-red tracks + red tracks only/total fibers] (*Figure 2G*). Results indicated that p21 depletion upregulated origin firing in the absence of stress (*Figure 2H*). Thus endogenous p21 is required for the optimal execution of the DNA replication program during unperturbed S phase.

## p21 low levels in cycling cells prevent replication stress in the absence of DNA damage

On the basis of the contribution of p21 to unperturbed DNA replication, we tested the effect of p21 depletion on the accumulation of markers of DNA replication stress. The intensity of γH2AX (*Mansilla et al., 2013*; *Ward and Chen, 2001*) increased both in U2OS transfected with p21 siRNA (*Figure 3A*) and in p21 −/− HCT116 cells (*Figure 3—figure supplement 1A–B*). Also, the number of cells with more than five 53BP1 foci (*Mansilla et al., 2013*; *Noon and Goodarzi, 2011*) (*Figure 3B* and *Figure 3—figure supplement 1C*) and the number of cells with RPA foci which reveals long stretches of single stranded DNA (*Bergoglio et al., 2013*; *Oakley and Patrick, 2010*) increased when p21 was depleted (*Figure 3C*).Thus, fork stalling and/or uncoupling events are more frequent in cells attempting to replicate DNA in the absence of p21 than in control samples.

Replication-associated defects could trigger the accumulation of perinuclear DNA or micronuclei (MN) in binucleated cells that have finished karyocinesis (*Fenech, 2000*). Notably, MN acumulated in cells transiently or permanently depleted from p21 (*Figure 3D* and *Figure 3—figure supplement 1D*, respectively). Another marker of genomic instability that is highly sensitive to replication defects is the rearrangements of common fragile sites (CFS) (*Le Tallec et al., 2014*; *Letessier et al., 2011*). Because of low origin density, the replication of CFSs is easily compromised by alterations in the replication program (*Letessier et al., 2011*; *Ozeri-Galai et al., 2011*, *2012*). Noteworthy, data in *Figure 3E and F* revealed that p21 depletion caused the accumulation of the FRA7H CFS instability to an extent similar to that caused by low doses of APH, a known inducer of CFS instability (*Bergoglio et al., 2013*; *Sutherland et al., 1985*). It follows that the depletion of p21 jeopardizes the duplication of hard-to-replicate genomic regions triggering replication-associated genomic instability.

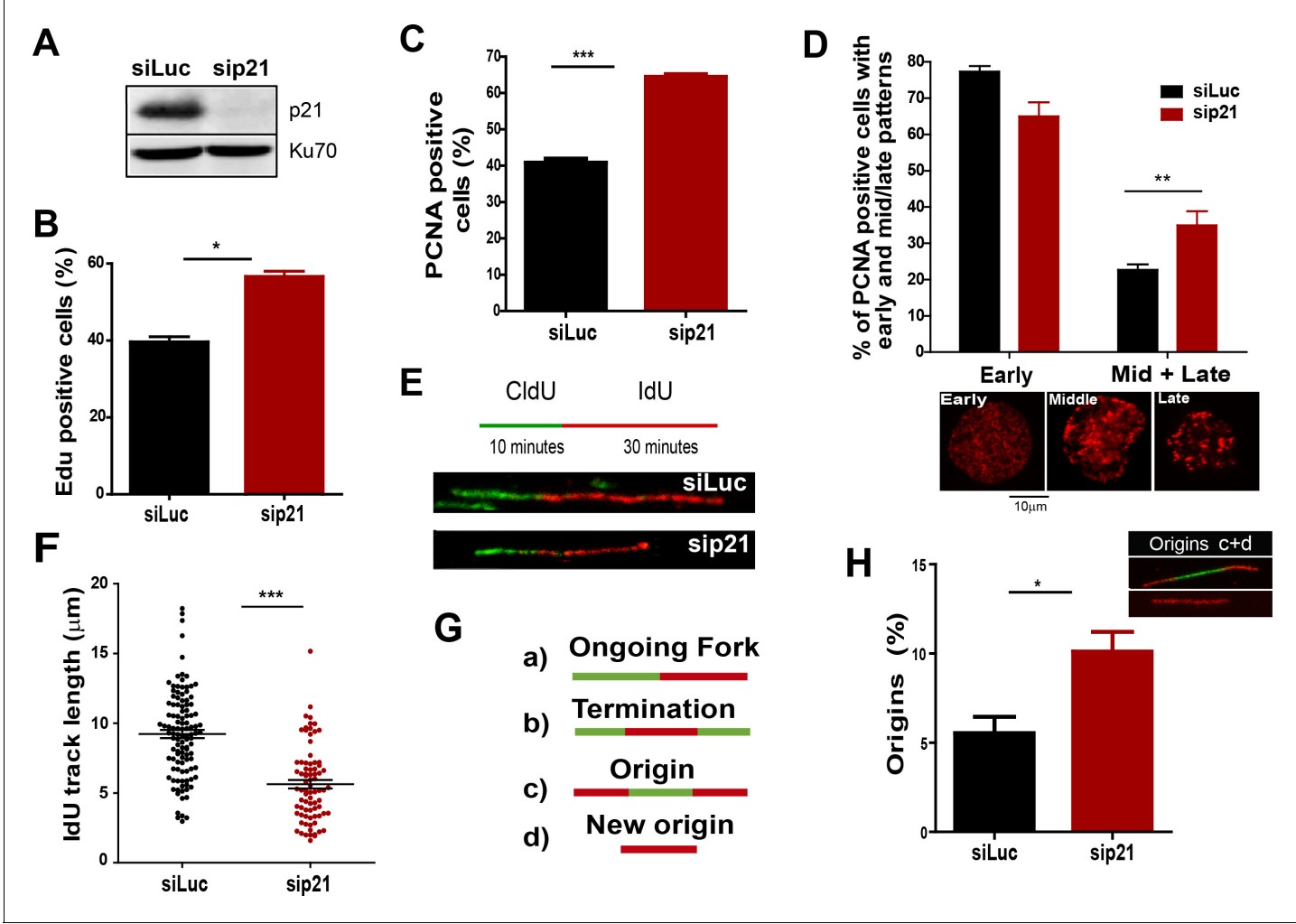

**Figure 2.** The depletion of endogenous p21 impairs the choreography of unperturbed DNA replication. (A) Western blot (WB) analysis showing p21 levels in U2OS cells transfected with the indicated siRNAs. (B) EdU positive cells. 200 nuclei/sample were analysed in three independent experiments. (C) The percentage of cells with CSK-resistant PCNA nuclear retention. 300 nuclei/sample were analysed in three independent experiments. (D) Relative amount of cells with early or mid/late PCNA distribution. 100 nuclei/sample were examined in three independent experiments. (E) Representative fibers from control (siLuc) or sip21 transfected cells. (F) IdU track length. 100 fibers/samples were analysed in three independent experiments. (G) Schematic representation of the different structures that can be measured in the fiber analysis. (H) Samples in F were used to analyse the frequency of origin firing as the relative number of origins [(red-green-red + red only fibers)/total fibers]. 200 fibers/samples were analysed in three independent experiments.

The following figure supplement is available for figure 2:

**Figure supplement 1.** Stable p21 depletion cause alterations in the DNA replication choreography of HCT116 cells.

## p21 prevents the aberrant use of the alternative DNA polymerase κ during the replication of undamaged DNA

Besides p21, novel negative regulators of alt DNA pols have been recently identified (*Bertolin et al., 2015*). One of them, USP1, has the ability to remove the ubiquitin moiety from PCNA (*Huang et al., 2006*; *Niimi et al., 2008*). Mono-ubiquitinated PCNA interacts with the UBM and UBZ domains of alternative DNA pols favouring their loading to the replisomes (*Bienko et al., 2005*; *Kannouche and Lehmann, 2004*; *Plosky et al., 2006*). During unperturbed replication, PCNA ubiquitination is limited by USP1 (*Huang et al., 2006*; *Niimi et al., 2008*), as evidenced by increased PCNA ubiquitination upon USP1 depletion (*Figure 4A and B*). In constrast, the level of PCNA ubiquitination was not modified when p21 was depleted (*Figure 4A*). Hence, p21 and USP1 regulate the

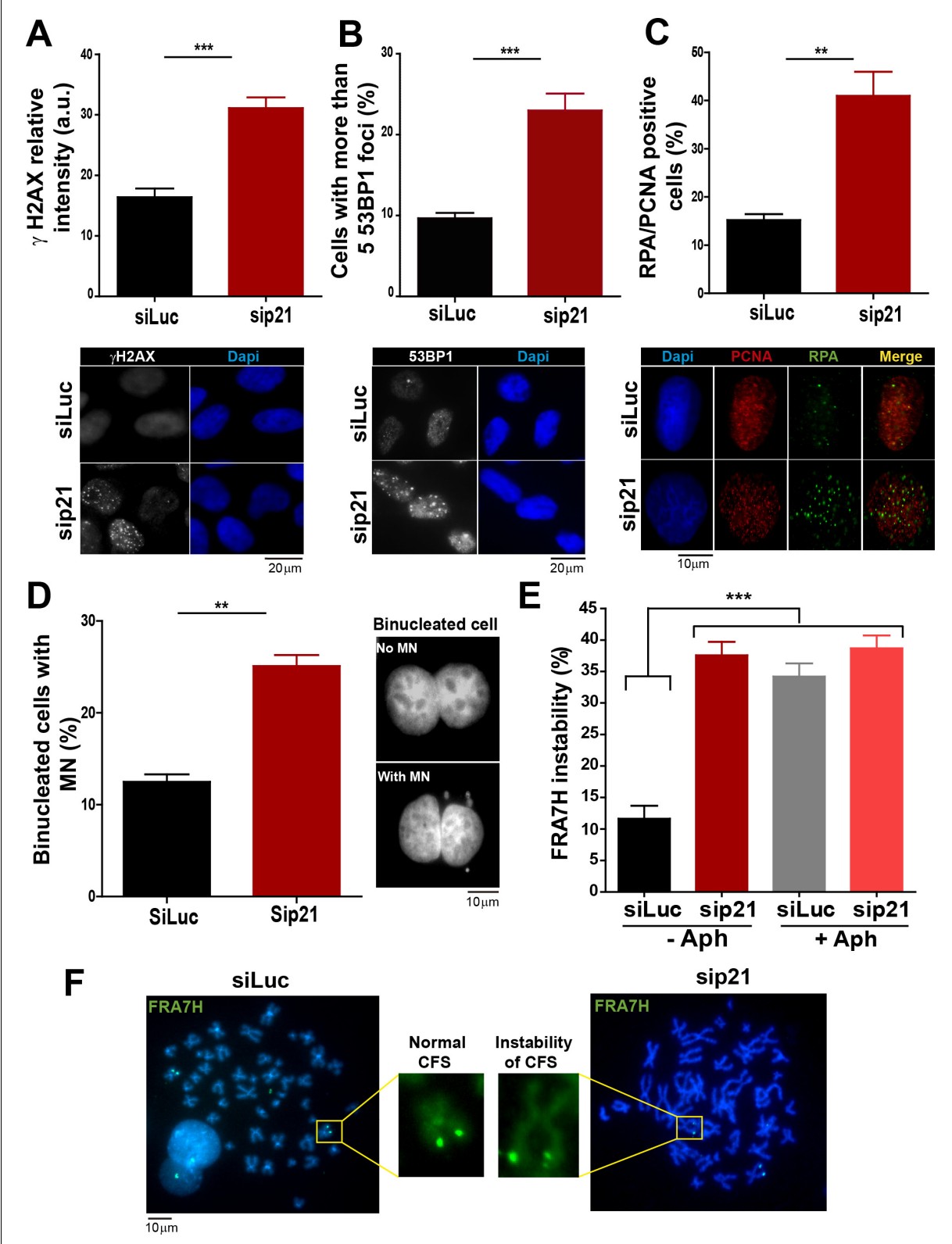

**Figure 3.** In the absence of DNA damage, replication stress markers and genomic instability increase when p21 is depleted. (**A**) Quantification of γ H2AX intensity in the nucleus of U2OS transfected with the indicated siRNA. 200 nuclei/sample were examined in three independent experiments. Representative images are shown in the lower part of the panel. (**B**) U2OS cells with more than five 53BP1 foci were quantified. 200 cells/sample were analysed in three independent experiments. Representative images are shown in the lower part of the panel. (**C**) Quantification of cells with RPA foci.

*Figure 3 continued on next page*

*Figure 3 continued*

150 nuclei positive for PCNA-resistance to CSK extraction/sample were examined in three independent experiments. (**D**) Quantification of cells with perinuclear DNA (MN) accumulation. 300 binucleated U2OS cells/samples were inspected in three independent experiments. Representative images are shown on the right. (**E**) Quantification of CFS expression for the indicated conditions. APH treatment corresponds to 0.2 μm for 24 hr. 50 metaphases from HCT116 cells/sample were examined in three independent experiments.

The following figure supplement is available for figure 3:

**Figure supplement 1.** Stable p21 depletion cause alterations in the genomic stability of HCT116 cells.

recruitment of alt pols by distinct mechanisms (*Figure 4C*). Intriguingly, despite such mechanistic differences, both p21 and USP1 depletion caused a similar effect (both in quality and extent) on nascent DNA elongation (*Figure 4D*), origin frequency (*Figure 4E*), the accumulation of cells with 53BP1 foci (*Figure 4F*) and the number of binucleated cells with MN (*Figure 4G*). These results reinforce the function of p21 as a negative regulator of alt DNA pols. Moreover, mechanistically distinct regulators of alt DNA pols are equally required to preserve DNA replication and genomic stability during unperturbed replication.

T. Huang and colleagues have previously reported that MN accumulation induced by USP1 depletion depends on Pol κ (*Jones et al., 2012*). Therefore, we set to explore the effect of p21 on Pol κ recruitment to DNA replication factories. First, we observed that Pol κ foci were formed only in a modest percentage of control cycling cells (siLuc in *Figure 5A–B*). However, when p21 was depleted, the percentage of cells with Pol κ foci raised significantly (*Figure 5A–B*). Second, the interaction of PCNA and GFP-Pol κ in the chromatin fraction increased when p21 was depleted (*Figure 5C*). Third, using PLA an increase in the number of endogenous PCNA/Pol κ interacting foci was revealed in p21-depleted samples (*Figure 5D–E*). We hypothesized that an increased recruitment of Pol κ to the replication forks in p21-depleted cell may slow down DNA elongation, triggering fork collapse and/or the generation of under-replicated DNA. To test this hypothesis, Pol κ was down-regulated in p21-depleted cells (*Figure 6A*) and different DNA replication parameters were tested. Forty eight hours after siRNA transfection, Pol κ depletion alone had no effect on most parameters, except from a modest increase in RPA foci formation (*Figure 6—figure supplement 1*). Such result may be in agreement with the role of Pol κ in the replication of non-B DNA regions such as G4 cuadruplex (*Betous et al., 2009*). Notably however, the simultaneous elimination of Pol κ and p21 prevented all the phenotypes associated with p21 depletion. Specifically, Pol κ depletion rescued the defective nascent DNA elongation (*Figure 6B*), the origin frequency (*Figure 6C*), the percentage of EdU positive cells (*Figure 6—figure supplement 1A*) and the increased number of cells with chromatin bound-PCNA (*Figure 6—figure supplement 1B*). Similarly, markers of replication stress such as RPA foci (*Figure 6—figure supplement 1C*), γH2AX (*Figure 6—figure supplement 1D*) and 53BP1 foci (*Figure 6D*) were downregulated after simultaneous depletion of p21 and Pol κ. Similar results were obtained when using a second siRNA specific for Pol κ in U2OS cells (*Figure 6—figure supplement 2A–D*) and when employing a different cell line, HCT116 p21 −/− cells (*Figure 6—figure supplement 2E–F*).

While MN accumulation was evident when p21 was knocked down, this was not observed after simultaneous depletion of Pol κ and p21 (*Figure 6E*). CFS instability modestly increased in Pol κ-depleted cells and more pronouncedly in p21-samples. Remarkably, in the context of p21 elimination, instead of increasing CFS instability Pol κ depletion reverted the instability caused by p21 depletion (*Figure 6F–G*) Collectively, these findings indicate that the misuse of Pol κ causes multiple alterations in the DNA replication of p21-deficient cells.

The recruitment of the alternative DNA polymerase pol eta (Pol η) to replication factories was also stimulated in the absence of p21 (*Figure 6—figure supplement 3A–C*). However, in contrast to Pol κ, Pol η depletion did not rescue the replication defects triggered by p21 depletion (see *Figure 6—figure supplement 3D–H*). Hence, the parameters of genomic instability explored in this study are predominantly associated with the misuse of Pol κ in p21-depleted samples.

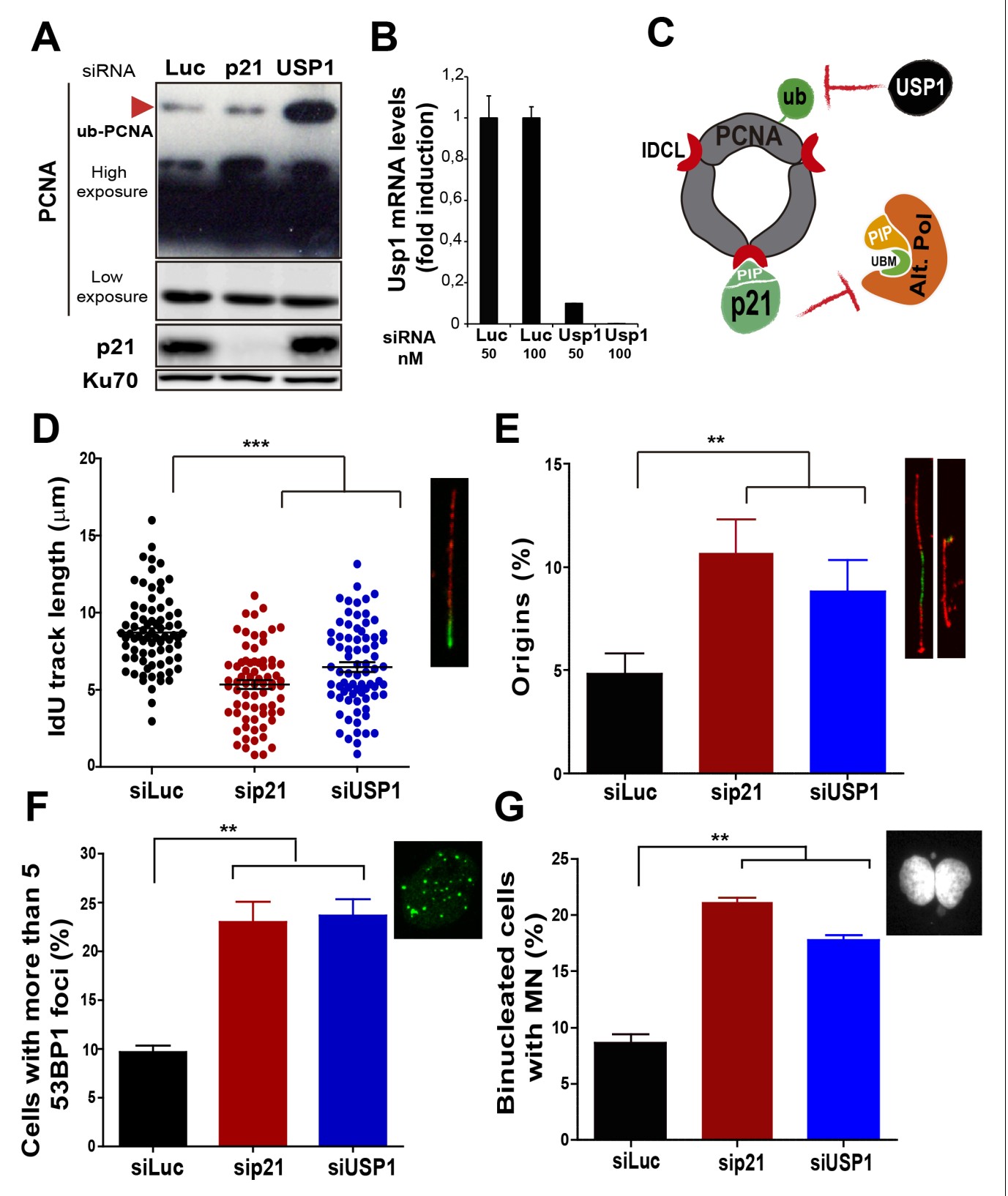

**Figure 4.** Mechanistically distinct regulators of alt DNA pols are required to facilitate unperturbed DNA replication. (**A**) W.B. analysis revealing PCNA, ubi-PCNA and p21 levels in U2OS cells transfected with the indicated siRNAs. (**B**) USP1 mRNA levels were examined by quantitative RT-PCR. (**C**) Model

*Figure 4 continued on next page*

*Figure 4 continued*

depicting the different mechanisms of PCNA regulation by p21 and USP1. (D) IdU track length was measured in 85 fibers/sample in three independent experiments. (E) Origin firing frequency. 150 fibers/sample were analysed in three independent experiments. (F) Quantification of cells with more than five 53BP1 foci. 200 U2OS cells/sample were analysed in three independent experiments. (G) MN accumulation. 300 binucleated cells/sample were analyzed in three independent experiments. Data on the effect of USP1 downregulation on the modulation of nascent DNA elongation (*Figure 4D*) and accumulation of cells with 53BP1 foci (*Figure 4F*) and micronuclei (*Figure 4G*) was reproduced from Jones et al, EMBO.

## p21 prevents mitotic transmission of DNA damage induced by Pol κ–dependent replicative stress

In addition to the well characterized and direct consequences of DNA replication stress, namely chromosomal breakage and aberrations, it has been shown that a fraction of under-replicated/unresolved genomic loci enter into mitosis. When not accurately processed in M phase, such DNA regions are converted into complex broken-DNA structures that are transmitted to daughter cells (*Minocherhomji et al., 2015*). In G1 phase such DNA structures are sequestered and shielded in nuclear compartments described as nuclear 53BP1 bodies (*Bergoglio et al., 2013*; *Harrigan et al., 2011*; *Lukas et al., 2011*). We therefore explored whether the altered DNA replication dynamics of p21-depleted cells could lead to the mitotic transmission of DNA damage. We first noticed an increase in the percentage of cells positive for the phosphorylated histone H3, a *bona-fide* marker of G2/M transition (*Minocherhomji et al., 2015*), upon p21 depletion which was again totally reversed by Pol κ depletion (*Figure 7A*). To confirm the persistence of under-replicated DNA in mitosis, we used previously reported protocols (*Federico et al., 2015*; *Minocherhomji et al., 2015*) to quantify 53BP1 body formation in G1 (EdU-negative). We found a significant increase in the number of spontaneous 53BP1 nuclear bodies in G1 in the absence of p21, as hallmark of incomplete DNA replication during the previous cell cycle, which was again dependent on Pol κ (*Figure 7B–D*). We conclude from these experiments that the choice of Pol κ in the absence of p21 is sufficient to increase the vulnerability of fragile genomic regions by delaying replication completion at these sequences. Such alteration of the DNA replication dynamics appears to be specific to Pol κ since they were not rescued when Pol η was depleted (*Figure 7E–F*).

## The PCNA-binding domain of p21 is necessary and sufficient to prevent the replication defects introduced by Pol κ

Having established that Pol κ triggers replication defects of p21-depleted cells, it was important to determine whether p21 prevents the loading of Pol κ to replication factories and if its ability to interact with PCNA is relevant to such function. We used the p21 mutants described in *Figure 1*, which are refractory to an siRNA directed to the 3′UTR of p21 (*Figure 8A*). Lentiviral transduction allowed the expression of p21 and p21^PIPMut* in almost all cells (*Figure 8B*). A fully functional PCNA binding domain in p21 was required to down-modulate Pol κ foci formation to control levels (*Figure 8C*). In agreement, the fork elongation defects and the excessive origin firing observed in p21-depleted samples were complemented by p21, but not by p21^PIPMut* (*Figure 8D–E*). Additionally, the key role of the PIP domain of p21 was supported by experiments performed in HCT116 p21−/− cells. Such experiments revealed that p21 but not the p21^PIPMut* mutant complement the replication defects of cells with a null mutation of the endogenous p21 alleles (*Figure 8 F–H*). Notably, the accumulation of markers of replication stress and genomic instability in p21-depleted cells was abolished when overexpressing p21, but not p21^PIPMut* (*Figure 8I–J*). Hence, we postulate a key role of the PIP domain of p21 in the preservation of DNA replication homeostasis.

## p21 preserves the genomic stability of primary cells

Given that our results indicate a novel antioncogenic role of p21 in the promotion of DNA replication it was important to determine whether this phenotype was not limited to cancer cells. To address such question we used primary cells from two independent sources: (a) human foreskin fibroblast (HFF) and (b) mesenchyimal stem cells isolated from umbilical cord (MSC). As shown in *Figure 9A and B*, transfection of p21 siRNA efficiently depleted p21 from both cell types. p21 elimination caused a reduction in the elongation of nascent DNA (*Figure 9A–C*), which was accompanied with

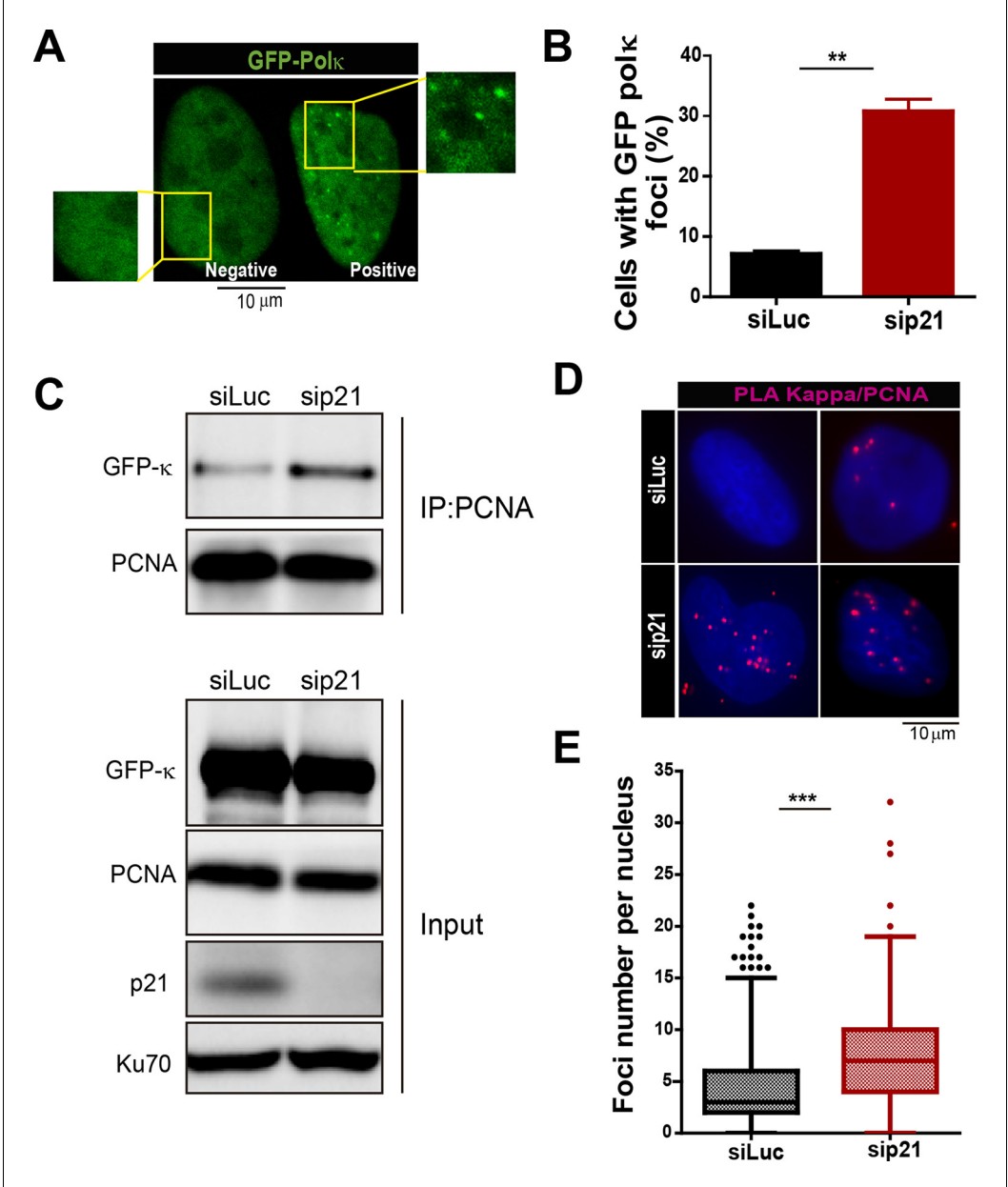

**Figure 5.** The recruitment of pol κ to replication-associated structures increases in the absence of p21. (A) U2OS cells were transfected with GFP-polκ. Representative images of cells with and without with GFP-pol κ foci. (B) Percentages of cells with GFP-pol κ focal organization. 150 nuclei/sample were analyzed in two independent experiments. (C) siLuc and sip21 depleted samples were subjected to chromatin immunoprecipitation using a monoclonal PCNA antibody. GFP-pol κ recruitment to chromatin was revealed by using GFP antibodies. The result was reproduced in three independent experiments. (D) Proximity Ligation Assay (PLA) between PCNA and endogenous pol κ was performed in U2OS cells. Two representative images of PLA in siLuc and sip21 cells are shown. (E) Quantification of PLA foci per nuclei. More than 1000 nuclei were analysed in three independent experiments.

an increase in origin firing (*Figure 9D*). In turn, such alteration in DNA replication parameters correlated with the accumulation of cells with 53BP1 foci (*Figure 9E–F*) and micronuclei (*Figure 9 G–I*). Hence, we postulate that p21 regulates protein-complex formation at the replisomes, promoting the choice of the most adequate polymerase and therefore protecting DNA replication homeostasis.

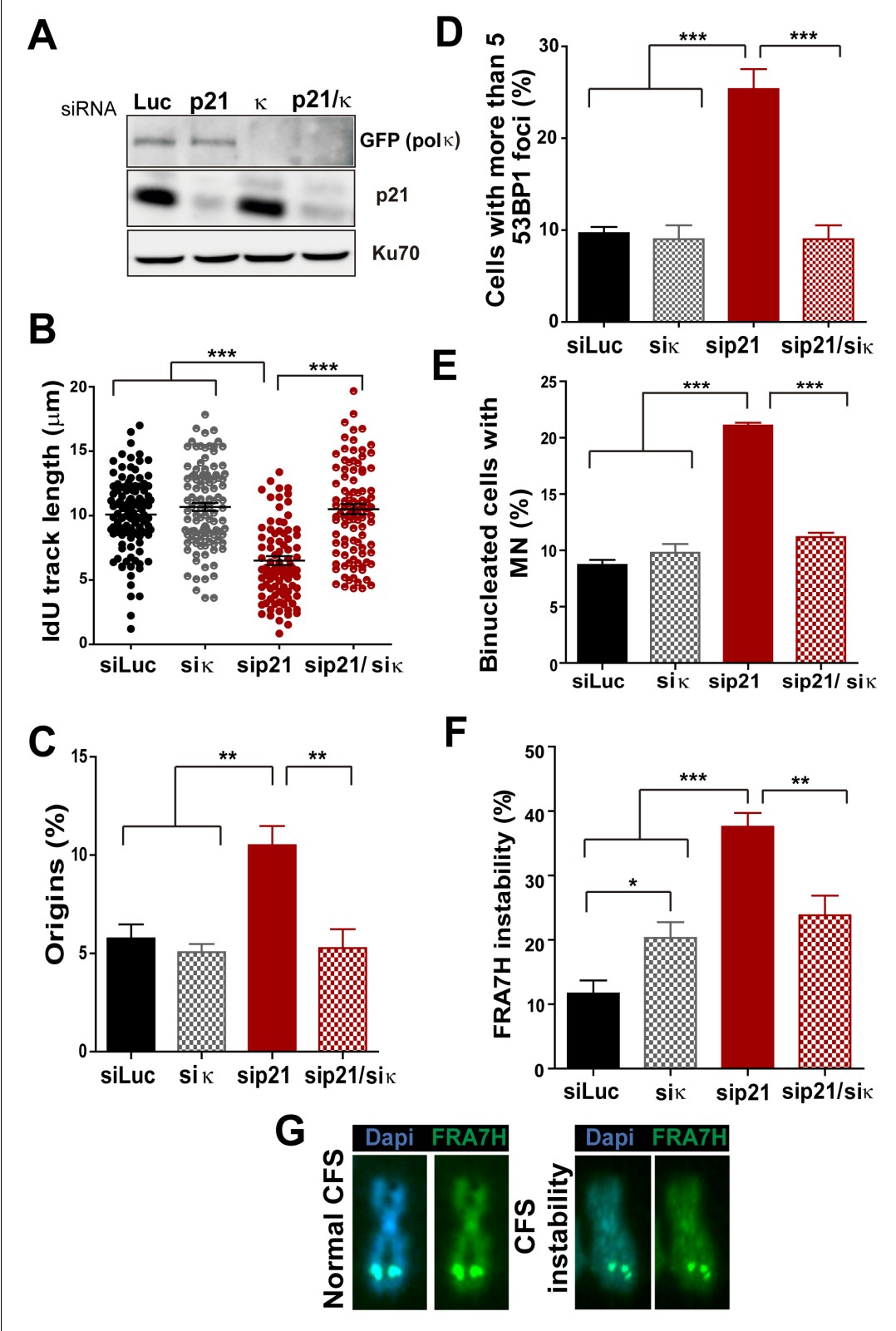

**Figure 6.** Pol κ depletion prevents the DNA replication defects and the genomic instability caused by p21 downmodulation. (A) Western blots showing GFP-pol κ and p21 levels in U2OS cells transfected with the indicated siRNAs. (B) Total IdU track length was evaluated in 100 fibers/sample in three independent experiments. (C) Frequency of origin firing. 200 fibers were analysed in three experiments. (D) Quantification of cells with 53BP1 foci. 200 cells were analysed in three independent experiments. (E) Quantification of MN accumulation. 300 binucleated cells/sample were analysed in three

*Figure 6 continued on next page*

*Figure 6 continued*

independent experiments. (**F**) Quantification of CFS instability in HCT116 cells transfected with the indicated siRNAs. 50 metaphases/sample were analysed in three independent experiments. (**G**) Representative images of the CFS analysed in F.

The following figure supplements are available for figure 6:

**Figure supplement 1.** Pol κ depletion prevents the accumulation of DNA replication stress markers caused by p21 downmodulation.

**Figure supplement 2.** Pol κ prevents the accumulation of DNA replication stress in p21 depleted cells independently of the siRNA used and in cells stably lacking p21.

**Figure supplement 3.** Pol κ but not pol η depletion prevents the DNA replication defects and the genomic instability caused by p21 downmodulation.

Such novel function of p21 depends exclusively on its ability to interact with PCNA and is needed at every S phase to ensure the accurate finalization of DNA replication (see model in *Figure 10*).

## Discussion

We describe a biologically relevant contribution of the interaction of p21 and PCNA in cells. We show that such interaction improves DNA replication dynamics since it is required to ensure the best rate of nascent DNA elongation. By promoting accurate DNA polymerase choice at the replisome, p21 acts as a tumor suppressor chronically at each duplication cycle, in the absence of exogenous sources of DNA damage.

### A novel function of p21 in the fine-tuning of DNA replication dynamics

An important implication of our findings is that the idea that p21 acts exclusively as a negative regulator of the cell cycle through the inhibition of cycling kinases (*Warfel and El-Deiry, 2013*) is simplified, incomplete and limited to specific DNA-damaging scenarios. The results presented herein conclusively show that p21 is more often a positive rather than a negative regulator of the cell cycle, as its contribution is required at every S phase. Remarkably, such contribution relies exclusively on the p21/PCNA interaction.

Previous work indicates that p21 can displace alt DNA pols from replicating DNA. First, Z. Livneh and colleagues showed that p21 impairs TLS events on transfected plasmids (*Avkin et al., 2006*). Second, lower levels of p21 in cycling cells must be eliminated to promote nascent DNA elongation across UV damage templates by alt DNA pols (*Mansilla et al., 2013*). Third, endogenous p21 delays the recruitment of alt DNA pols to UV-damaged replication factories in a manner that correlates with the extent of p21 degradation (*Mansilla et al., 2013*; *Soria et al., 2008*). Fourth, the PCNA-binding domain of p21 is required to inhibit TLS activation (*Avkin et al., 2006*; *Mansilla et al., 2013*; *Soria et al., 2008*). It follows that alt DNA pols might be selectively sensitive to p21 levels which are insufficient to inhibit CDKs (*Soria and Gottifredi, 2010*). Such a difference in the levels of p21 required to inhibit different cellular processes indicates that p21 is locally concentrated at, and/or has very high affinity for PCNA (*Soria and Gottifredi, 2010*). Both replicative and alt DNA pols bind the IDCL of PCNA through PIR or PIP (PCNA interacting protein) regions (*Bertolin et al., 2015*; *Moldovan et al., 2007*). The PIR of p21 binds PCNA strongly, so that p21 is capable of disrupting the weaker interactions between PCNA and alt DNA pols (*Hishiki et al., 2009*; *Mansilla et al., 2013*; *Tsanov et al., 2014*) without affecting replicative DNA pols which have multiple PIP domains (*Soria and Gottifredi, 2010*; *Soria et al., 2008*). Such difference may relate to the fact that replicative DNA pols utilize multiple domains and different sites to interact with PCNA (*Johansson et al., 2004*; *Moldovan et al., 2007*).

There is a tight cross-regulation of PIP box-containing proteins at replication forks which is not yet completely understood. On the one hand, it has been demonstrated that proteins with strong PIP boxes such as p21 can remove alt pols such as Pol κ from replication factories (this report and [*Tsanov et al., 2014*]). On the other hand, this and previous reports (*Jones et al., 2012*; *Pillaire et al., 2007*) suggest that the alt Pol κ can displace replicative pols from replisomes. Conversely however, p21-PCNA interaction in S phase does not impair genomic DNA synthesis

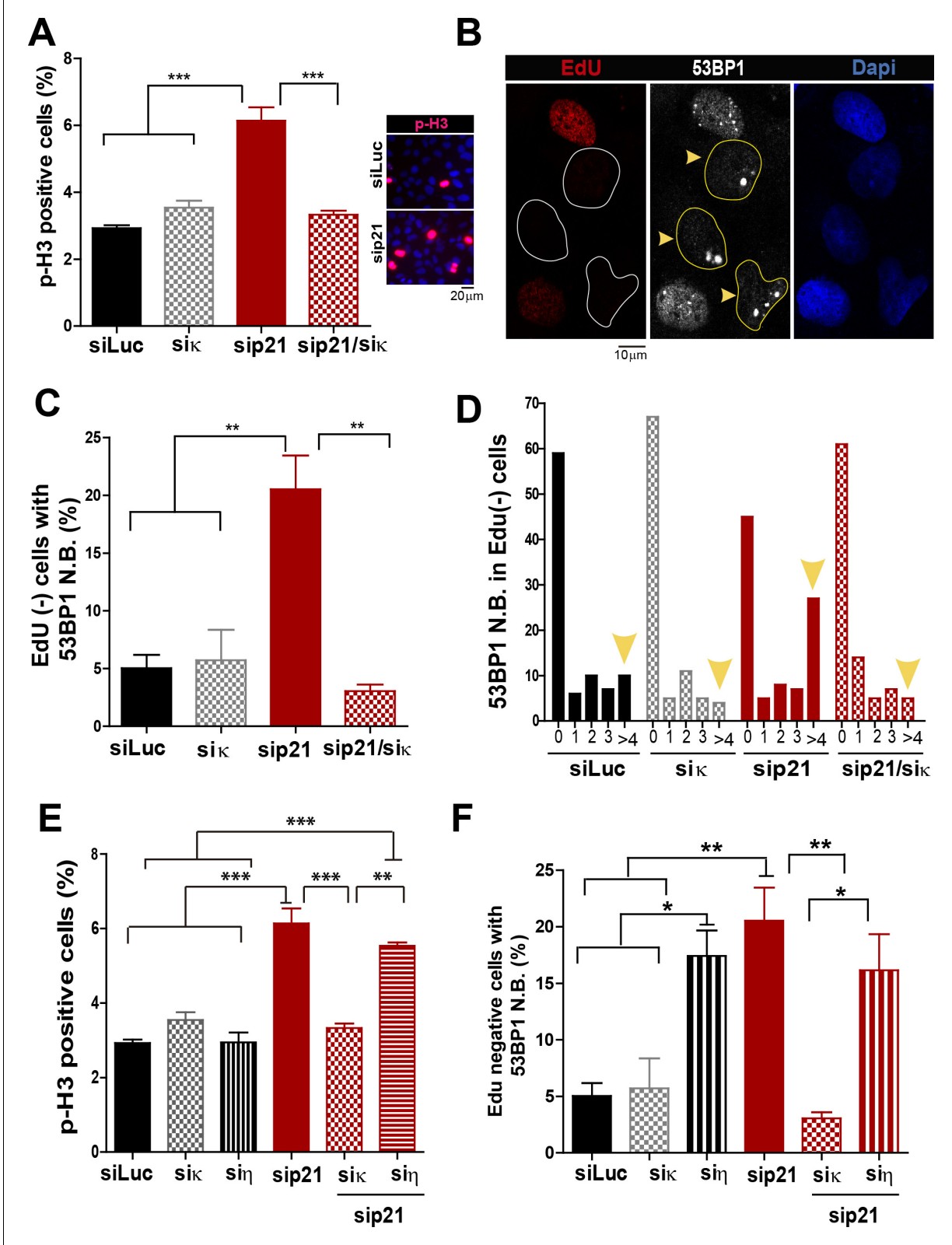

**Figure 7.** Pol κ-mediated replication defects of p21-depleted cells are transmitted to the M and G1 phases of the cell cycle. (A) Quantification of phospho-H3 positive U2OS. 200 cells/sample were analysed in three independent experiments. (B) Representative images of 53BP1 bodies outside S phase. Yellow arrows indicate EdU negative cells with 53BP1 foci (C) Percentages of EdU negative cells which are positive for 53BP1 bodies. 200 nuclei/sample were analysed in three independent experiments. (D) Distribution of EdU negative cells with increasing number of 53BP1 bodies per cell in the

*Figure 7 continued on next page*

*Figure 7 continued*

experiments showed in C. 100 nuclei/sample were analysed in three independent experiments. (E) Phospho-H3 accumulation was quantified in 200 nuclei/sample of three independent experiments. Samples depleted from pol κ and pol η were compared. (F) Percentages of cells with more than five 53BP1 foci were determined after analysing 200 nuclei/sample in three independent experiments. Samples depleted from pol κ and pol η were compared.

(*Soria et al., 2008*) suggesting that p21 cannot displace replicative DNA pols from the replisome. Hence, some yet unknown factors/events need to be identified to understand the hierarchy of PCNA partners at forks. Moreover, variables such as the sequence of the DNA which is being replicated and the average retention time of a PCNA partner at different regions of the chromatin might have a role in such a puzzling cross-regulation among PCNA partners. We have shown that Pol κ is required for checkpoint activation at stalled forks (*Betous et al., 2013*) and during the replication of repetitive sequences (*Hile et al., 2012*) or naturally occurring non-B DNA structures (*Betous et al., 2009*). In agreement, Pol κ depletion affected the CFS expression and RPA accumulation in our experimental settings. It is therefore possible that certain DNA regions may benefit from Pol κ-dependent DNA synthesis (*Betous et al., 2009*) while others it might require active displacement of Pol κ by p21. If that is the case, p21 may prevent Pol κ recruitment to specific regions of the genome while prompt p21 degradation by CLR4$^{Cdt2}$ might rapidly allow Pol κ binding to others. Such p21 function could be executed either by promoting the dissociation of Pol κ from replisomes or by preventing its recruitment to DNA regions that must be necessarily copied by replicative DNA pols (such as for example B-regions in the exonic DNA).

## The complex regulation of Pol κ by p21

The PIR domain of p21 prevents the recruitment of Pol η, Pol ι, Pol κ and Rev1 to UV-damaged replication factories (*Bertolin et al., 2015*; *Mansilla et al., 2013*). Hence, it is unlikely that during unperturbed replication p21 would act as a specific inhibitor of Pol κ. The same is valid for USP1 because the modulation of PCNA ubiquitination should regulate all alt DNA pols (*Huang and D'Andrea, 2006*; *Jones et al., 2012*). Hence, p21 and USP1 depletion may result in increased loading of all alt DNA pols to undamaged DNA. In fact, others have reported increased spontaneous mutation frequency in the hypoxanthine phosphoribosyl transferase (hprt) locus of p21−/− cells, (*McDonald et al., 1996*). Such defects might depend on the unleashed activity of alt DNA pols other than Pol κ (*Yang and Woodgate, 2007*). Notwithstanding this, the phenotypes described herein are intimately associated with processive DNA synthesis events, and Pol κ is a very processive alt DNA pol (the most processive in the Y family) (*Ohashi et al., 2000*; *Zhang et al., 2000*). Because Polκ is less processive than replicative DNA pols (*McCulloch and Kunkel, 2008*; *Ohashi et al., 2000*), the coupling of a replicative DNA pol and Pol κ at a single replisome may most likely generate asynchronic speed in both DNA strands. In turn, this may cause the transient accumulation of ssDNA in one strand (see *Figure 3C*). In fact, low doses of APH, which are known to disrupt the DNA pol homeostasis during DNA replication, cause similar levels of CFS expression as p21 depletion (see *Figure 3E*). Hence, the local degradation of p21 by CLR4$^{CDT2}$ may maintain the most efficient DNA elongation speed by allowing the rapid and dynamic exchange of replicative DNA pols for Pol κ. In fact, a minor shift in the timing of CLR4$^{CDT2}$-dependent p21 degradation is sufficient to accumulate cells in S phase (*Coleman et al., 2015*). Notably, the elimination of not only p21 but also USP1 (*Jones et al., 2012*) increases the use of Pol κ during undamaged DNA replication, suggesting that the slightest modulation of Pol κ activity perturbs the DNA replication dynamics. In agreement, the expression of a Pol κ mutant with increased affinity for PCNA and the overexpression of Pol κ generate genomic instability and tumor formation (*Bavoux et al., 2005a*; *2005b*; *Bergoglio et al., 2002*; *Hoffmann and Cazaux, 2010*; *O-Wang et al., 2001*; *Jones et al., 2012*).

## The biological consequences of excessive genomic DNA synthesis by Pol κ

It is relevant to mention that the p21 knockout mice develop spontaneous tumors at 16 months (*Martin-Caballero et al., 2001*), suggesting that modest but accumulative defects eventually trigger oncogenic transformation. Such observation highlights a role of p21 in the face of chronic, rather

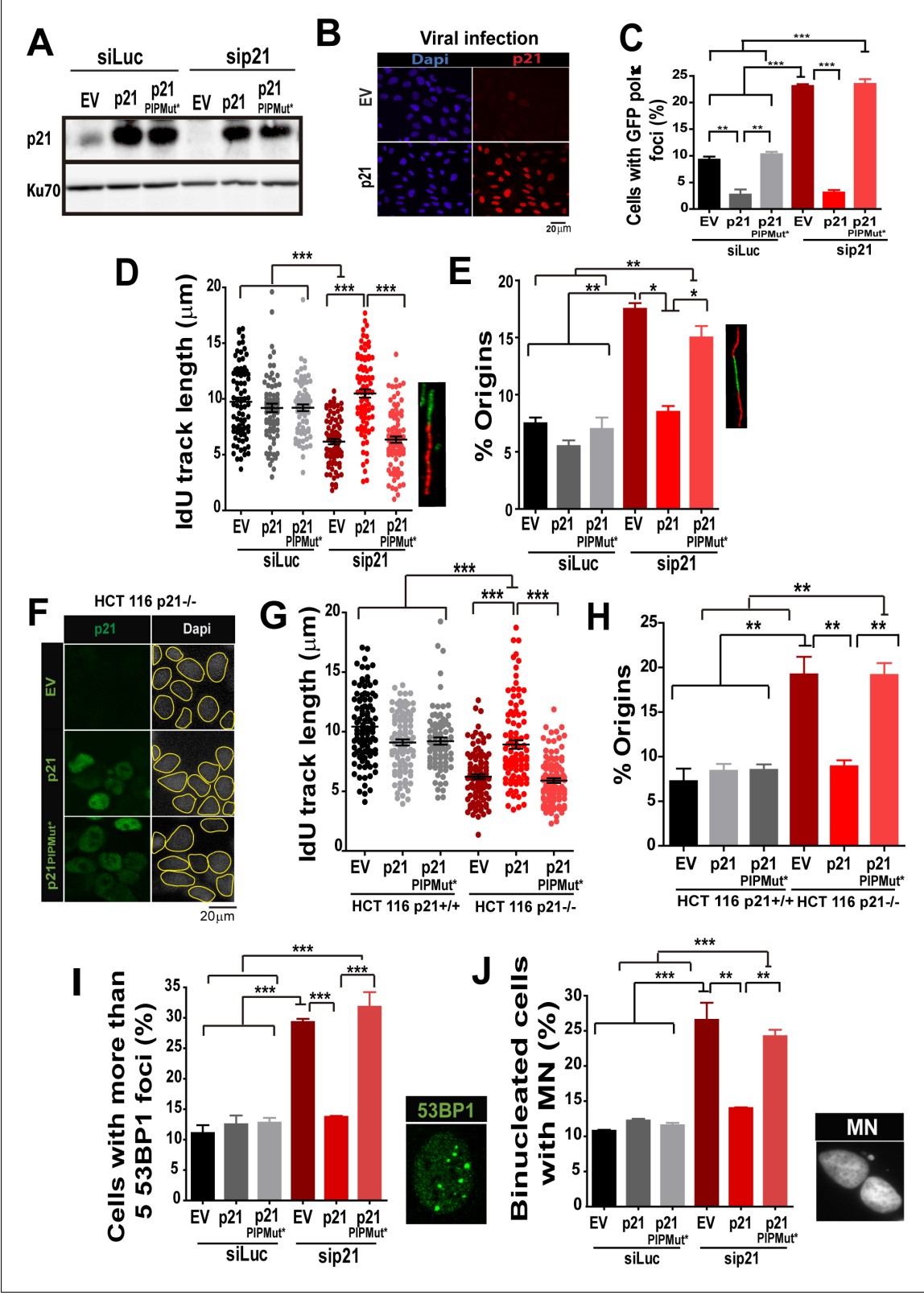

**Figure 8.** The PIR domain of p21 and not the CDK-binding domain is required to prevent DNA replication defects in p21 depleted cells during the unperturbed S phase. (**A**) W.B. analysis was performed to evaluate p21 expression in U2OS cells transfected with control or p21 siRNA and infected with p21 or p21$^{PIPMut*}$ lentiviruses 6 hr later. (**B**) Representative panel showing the efficiency of lentiviral infection (>90%). (**C**) GFP-pol κ focal organization in samples depleted from p21 and infected with p21 or p21$^{PIPMut*}$. (**D**) IdU track length was measured in 100 fibers/sample in three

*Figure 8 continued on next page*

*Figure 8 continued*

independent experiments. (E) Frequency of origin firing. 100 fibers/sample were analysed in three experiments. (F) Infection were performed in HCT116 p21+/+ and p21−/− cells. Representative panel showing the efficiency of lentiviral infection in HCT116 p21−/− s. (G) IdU track length was measured in 100 fibers/sample in three independent experiments. (H) Frequency of origin firing was measured in the experiments shown in G. 200 fibers/sample were analysed. (I) Quantification of the percentage of cells with more than 5 53BP1 foci/cell. 200 cells/sample were evaluated in three independent experiments. (J) Quantification of MN accumulation. 200 binucleated cells were analysed in three independent experiments.

than acute stress and may correlate with the multiple defects in DNA replication revealed when depleting p21 from primary cells (*Figure 9*). While p21 is a negative regulator of TLS, its role in undamaged DNA replication (revealed in this manuscript) is most possibly *unrelated* to TLS inhibition. In fact, if p21 depletion caused the accumulation of DNA lesions, the elimination of an alt polymerase should: (a) aggravate the defect (if the alt pol is essential to promote DNA replication across the replication barrier) or (b) not alter such defect (if the replication across such barrier is also achieved by another alt polymerase) (*Bertolin et al., 2015*). As our data show that the elimination of Pol κ promotes DNA replication, it does not support a model in which p21 depletion generates substrates (DNA lesions) for alt pols. On the contrary, it suggests that undamaged DNA is being misused by the alternative polymerase pol κ as a replication template. Hence, the contribution of p21 to unperturbed-DNA replication is most probably related to the control of DNA synthesis events on undamaged templates. In such scenario, the colocalization of the p21-CLR4$^{cdt2}$ complex at replication forks may be crucial to control the composition of the active replisome, allowing the selection of the most adequate polymerase 'on the go'. Such dynamic regulation would favour the most effective DNA replication speed allowing the timely activation of late and hard-to-replicate genomic regions. The coupling of p21 with ongoing replisomes may also favor other p21 functions as homologous recombination (HR) events. While such p21 function has been linked to CDK inhibition (*Mauro et al., 2012*), it is still possible that the loading of p21 to PCNA serves to orchestrate multiple transactions at the elongating DNA. While more work is required to reveal yet hidden cross-regulations among members of the replisome, our work unveils a novel tumor suppressor function of p21 dependent on its ability to interact with PCNA. Such a role of p21 is central for genome maintenance in the absence of DNA damage at every S phase and which might be key to prevent spontaneous oncogenic events.

## Materials and methods

### Cell culture and reagents

All cell lines were received from reliable sources, either public repositories or directly, from the laboratory that generated them. They were never frozen at passage higher than eight. U2OS (source: ATCC), and HCT116 p21+/+ and HCT116 p21−/− (gifts from B. Vogelstein-Johns Hopkins University, Baltimore) were received directly from ATCC and Johns Hopkins University respectively. Samples were minimally amplified and frozen after the reception and were used for limited passages after thawing of those primary stocks. We have verified the identity of the cell lines in terms of the pathways which are key for this study. We have verified the p53 and p21 status by RT-PCR. The three cell lines are positive for p53 and only U2OS and HCT 116 p21+/+ were positive for p21, while HCT 116 p21−/− was not. None of the used cell lines were in the list of commonly misidentified cell lines maintained by the International Cell Line Authentication Committee (*Capes-Davis et al., 2010*) updated in http://iclac.org/databases/cross-contaminations/). Samples were grown in Dulbecco's modified Eagle's medium (Invitrogen, Cambridge, Massachusetts) with 10% fetal calf serum in a 5% CO2 atmosphere. Samples are routinely tested for mycoplasma by PCR every two weeks.

Umbilical cord mesenchymal stem cells (UC-MSC) were isolated from Wharton jelly tissues. Small pieces of the umbilical cord, excluding the major vessels, were layered onto plastic culture plates and cultured in D-MEM supplemented with 10% SFB and penicillin-streptomycin. After 14 days, cells started to emerge from the tissue pieces. When plates were almost confluent, umbilical cord pieces were removed and cells trypsinized and frozen. Multilineage differentiation potential of UC-MSC was assessed using StemPro Adipogenesis, Osteogenesis or Chondrogenesis Kit (Life Technologies,

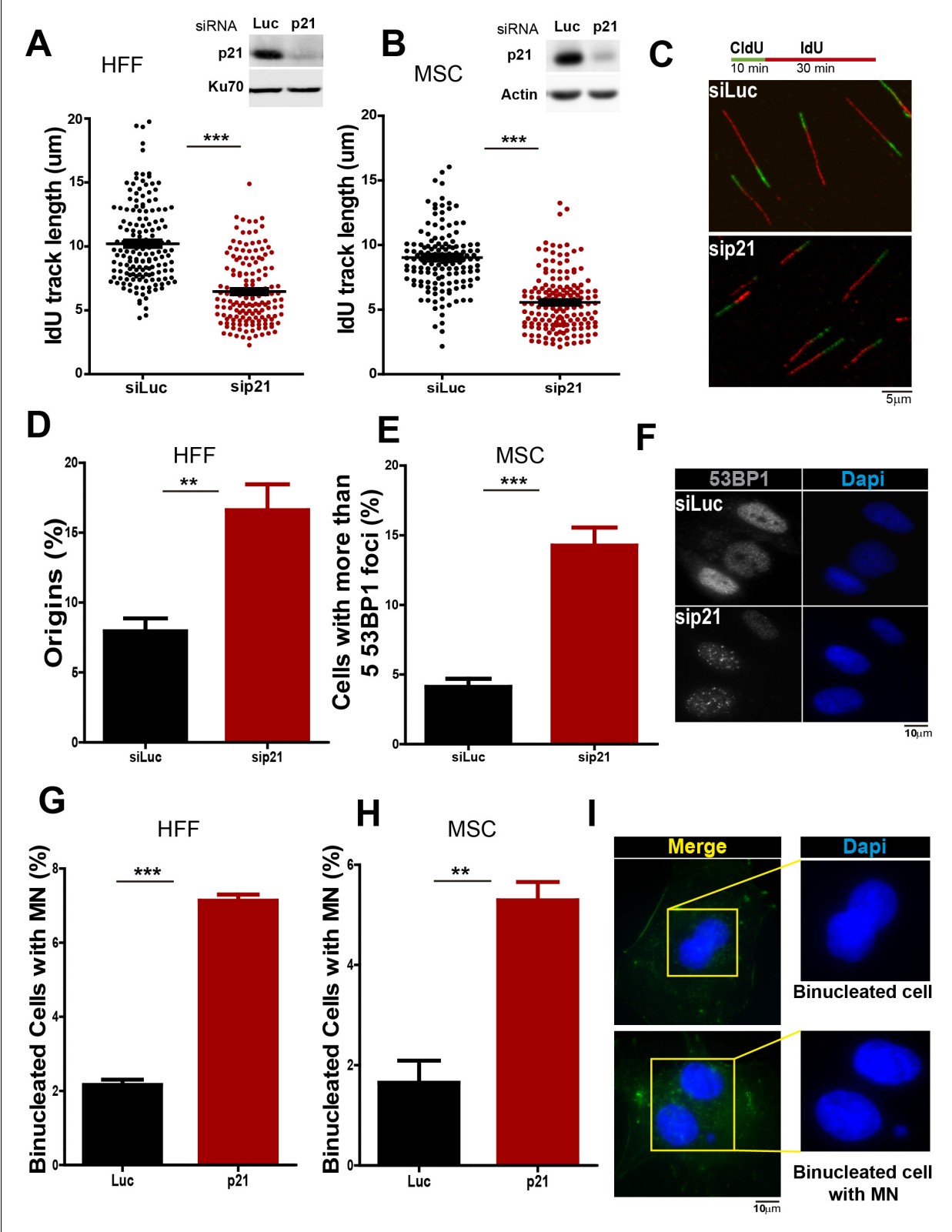

**Figure 9.** p21 depletion promotes replication stress and genomic instability in primary human cells. (**A** and **B**) IdU track length in Human Foreskin fibroblasts (HFF) and Umbilical Cord Mesenchymal Stem cells (MSC). 100 fiber/sample were analysed in three independent experiments. Western blot showing p21 depletion in the indicated cell line. (**C**) Representative fibers in siLuc and sip21 HFF cells. (**D**) Origin frequency in HFF cells. 200 fiber/sample were analyzed in three independent experiments. (**E**) Cells with more than 5 53BP1 foci were analyzed in MSC cells. 200 nuclei/sample were

*Figure 9 continued on next page*

*Figure 9 continued*

analyzed in three independent experiments. (**F**) Representative images of 53BP1 in siLuc and sip21 MSC cells. (**G** and **H**) Quantification of MN accumulation. 200 binucleated cells were analyzed in HFF and MSC cells respectively in three independent experiments. (**I**) Representative images of Binulceated cells with and without MN.

US) as per instruction of the manufacturer. Human Foreskin Fibroblasts (HFF) were obtained under informed consent from his parents from the foreskin of a three-year old boy undergoing a scheduled surgery. HFF were obtained using a similar protocol to the one used with UC-MSC. Both cell lines were grown in DMEM supplemented with 10% FBS and antibiotics up to a 15 passage. All isolations of primary cells were approved by FLENI Ethic Committee after reviewing the research protocol (*Luzzani et al., 2015*).

The HFF (Human Foreskin Fibroblasts) were analyzed by karyotyping to assess genomic integrity after 10 passages. Experiments were performed using cells up to passage number 15 to prevent the occurrence of abnormalities. In addition, cells surface markers are periodically analized by flow citometry. MSC line were used up to passage number 10 to prevent genomic aberrations and lose of stem cells´ properties. MSC surface marker expression profile and their differentiation potential were

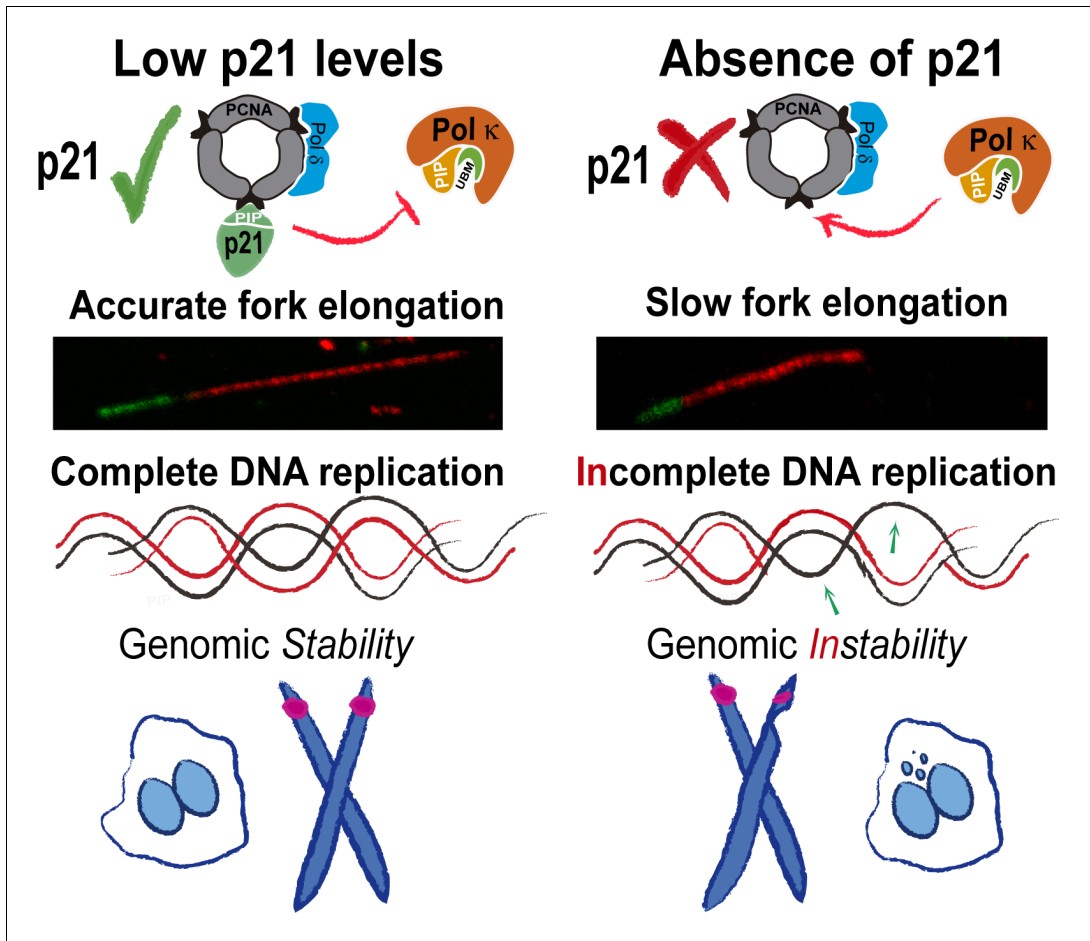

**Figure 10.** Model depicting the implication of our findings. p21 levels in S phase are low but sufficient to prevent pol κ loading to replication forks. When p21 is absent, the abnormal use of pol κ during unperturbed replication impairs nascent DNA elongation. While origin firing increases to compensate the slow fork progression, late replicating and origin-poor DNA regions, that is. CFS, are inefficiently duplicated. The replication defects accumulated in p21-depleted samples lead to genomic instability and transmission of replication defects to G1.

periodically analized by flow citometry and by chondregic, adipogenic and osteogenic differentiation, respectively. MSC immunomodulatory properties were also assessed by CFSE lymphocyte proliferation assay. All cell lines were weekly analyzed for mycoplasm contamination by PCR. Transfections of siRNAs were performed using Lipofectamine RNAiMAX (Thermofisher) and Jet Prime (Polyplus, France) was used when delivering both siRNAs and plasmid expression vectors.

## siRNAs, vector expression plasmids and lentiviral infection

GFP-Pol η and GFP-Pol κ were gifts from Dr. A Lehmann (*Kannouche et al., 2001*). GFP-PCNA was kindly provided by Dr. M C Cardoso (Max Delbrück Center for Molecular Medicine, Berlin, Germany) (*Leonhardt et al., 2000*) siRNAs were purchased from Dharmacon (Lafayette , Colorado):

siRNA p21 3'UTR:GGAACAAGGAGUCAGACAU;
siRNAUSP1:UCGGCAAUACUUGCUAUCUUA (*Jones et al., 2012*);
siRNA Pol κ1: AAGAUUAUGAAGCCCAUCCAA (*Vallerga et al., 2015*);
siRNA Pol κ2: AACCUCUAGAAAUGUCUCAUA (*Jones et al., 2012*)
siRNA Pol η: CUGGUUGUGAGCAUUCGUGUA (*Vallerga et al., 2015*).

In this work, we used a previously described p21 expression vectors (CS2-p21) and generated the p21$^{PIPMut*}$ (CS2-p21 with the M147 to A, D149 to A and F150 to A mutations which were shown to disrupt the PCNA/p21 interaction [*Gulbis et al., 1996*; *Mansilla et al., 2013*]) Both the p21 and the p21$^{PIPMut*}$ vectors bear three point mutations in its CDK binding domain (W49 to R,F51 to S and D52 to A) which disrupt the CDK2/p21 interaction (*Gulbis et al., 1996*; *Mansilla et al., 2013*). Primers used to generate p21 mutations were described and validated in our previous work (*Mansilla et al., 2013*) and both mutations were characterized previously (*Gulbis et al., 1996*).

p21 and p21$^{PIPMut*}$ were cloned into the lentiviral transfer plasmid pLenti CMV/TO Puro (Plasmid #17482, Addgene, 3rd generation) using BamHI and XbaI. Packaging and envelope 2nd generation vectors used were psPAX2 and PMD2.G respectively. Lentiviral particles were obtained by transfecting with Jet prime Hek293T cells in 60 mm dishes in a 5:5:1 ratio (pLenti: psPAX2:PMD2). 48 hr after transfection lentiviral particles were collected from the supernatant of Hek293T cells, centrifuged, filtered with a 0.45 μm filter and stored at −80°C. Lentiviral preparations were slowly thawed and used to infect U2OS cells in 24/12 well dishes in the presence of 8 μg/ml of polybrene. Infection efficiency was approximately 90% (*Figure 7B*).

## Immunostaining and fluorescence detection

For the quantification and inmunodetection of specialized GFP tagged Pol κ and Pol η, 53BP1 and γH2AX respectively, cells were fixed in 2% paraformaldehyde (PFA)/2% sucrose and permeablized with 0.1% Triton X-100 in phosphate buffered saline (PBS) as described previously (*Mansilla et al., 2013*). For the detection of chromatin-bound protein ice cold, 0.5% Triton CSK buffer was used (10 mM Pipes, pH 7.5, 100 mM NaCl, 300 mM sucrose, 3 mM MgCl2). Proteins were preextracted 1 min or 5 min when detecting p21 or RPA/PCNA respectively. EdU was detected following manufacturer's instructions (Click-iT EdU kit– C10338 Invitrogen). Blocking was performed overnight in PBS 2% donkey serum (Sigma, St. Louis, Missouri). Coverslips were incubated for 1 hr in primary antibodies: 53BP1(Santa Cruz, Dallas, Texas), γH2AX (EMD Millipore, Massachusetts), p21 (c-19 Santa Cruz), RPA (NA18 EMD Millipore), PCNA (Abcam), pH3 (ser10 Millipore). Secondary anti-mouse/rabbit-conjugated Cy2/Cy3 antibodies were from Jackson Immuno Research and anti-rabbit alexa 488 from Invitrogen. GFP-tagged specialized Y polymerases and GFP-PCNA were detected by GFP autofluorescence. Nuclei were stained with DAPI (SIGMA). Images were obtained with a Zeiss Axioplan confocal microscope or a Zeiss Axio Imager A2.

## Protein analysis

For Western blot analysis, samples were lysed in Laemmli buffer. Antibodies used were anti-p21 (c-19; Santa Cruz Biotechnology) anti-polη (H-300; Santa Cruz Biotechnology), anti-KU70 (A9; Santa Cruz Biotechnology), and anti-GFP (Santa Cruz Biotechnology). Secondary antibodies (Sigma) and ECL detection (Amersham GE Healthcare, Milwaukee, Wisconsin) were used according to the manufacturers' instructions. Western blot images were taken with Image QuantLAS4000 (GE Healthcare), which allows capture and quantification of images within a linear range. These images were then quantified with the ImageJ software when indicated.

## DNA fiber spreading

DNA fibers were analysed using a protocol previously used by us (*Mansilla et al., 2013*) with a minor change in the time of labelling. Exponentially growing cells were pulse labeled with CldU (20 µM) for 10 min, washed twice,and incubated with IdU(200 µM) for additional 30 min. Cells were trypsinized and lysed with 6 µl of 0.5% SDS, 200 mM Tris–HCL (pH 7.4) and 50 mM EDTA buffer onto clean glass slides, which were tilted, allowing DNA to unwind. Samples were fixed in 3:1 methanol/acetic acid and denatured with HCL (2.5 N) for 1 h, blocked in PBS 5%Bovine serum albumin (BSA) for 15 min and incubated with the mouse anti-BrdU (Becton Dickinson, USA) to detect IdU, donkey anti-mouse Cy3-conjugated secondary antibody (Jackson Immuno Research, West Grove, Pennsylvania), rat anti-BrdU (Accurate Chemicals, Westbury, New York) to detect CldU and donkey antirat Alexa 488 secondary antibody (Invitrogen). Slides were mounted with Mowiol 488 (Calbiochem), and DNA fibers were visualized using a Zeiss Axioplanconfocal microscope. Images were analysed using Zeiss LSM Image Browser software and Image J software. Each data set is derived from measurement of 85–100 fibers.

## Chromatin immunoprecipitation of PCNA

Chromatin immunoprecipitations were performed as described in (*Soria et al., 2008*). U2OS cells were plated in 100 mm culture dishes, transfected with either Luc or p21 siRNA and 24 hr later transfected with GFP-Pol Kappa. Cells were rinsed twice with cold PBS and then extracted with 5 mL of CSK buffer (250 mM sucrose, 25 mM KCl, 10 mM HEPES at pH 8.0, 1 mM EGTA, and 1 mM MgCl2) for 10 min in ice. The CSK-extracted cells were fixed with 1% formaldehyde in PBS (4.5 mL) for 12 min. Then, 0.5 mL of 1 M glycine was added for 5 min to quench the cross-linking reaction. The cross-linked nuclei were rinsed with PBS and then lysed in 750 µL of IP lysis buffer (10 mM Tris·HCl at pH 7.5, 25 mM FNa, 20 mM NaCl, 1% Nonidet P-40, 1% Na-Deoxicholate, and 0.1% SDS) freshly supplemented with protease and phosphatase inhibitors. Lysates were scraped from the plates and transferred into 1.5-mL Microfuge tubes. Samples were sonicated (Bioruptor Sonication System, Diagenode) and clarified by centrifugation at 12,000 × g for 30 min at 4°C. PCNA was immunoprecipitated overnight at 4°C with 5 µL of monoclonal PCNA antibody (PC-10 AC; Santa Cruz Biotechnology). Samples were washed, lysed in Laemmli buffer and heated at 99°C for 30 min to revert the crosslink, and resolved in SDS/PAGE for Western blot analysis.

## Proximity ligation assay (PLA)

U2OS cells were seeded onto 22 mm x 22 mm coverslips in 6 well plates. 24 hr later cells were pre-extracted with ice cold CSK buffer (PIPES 10 mM, NaCl 100 mM, sucrose 300 mM, MgCl2 3 mM,triton 0.5%) for 2 min and fixed with PFA 4% for 10 min. After pre-extraction and fixation, interaction between endogenous p21 and PCNA was detected following manufacturer's instructions Duolink In Situ – Fluorescence (SIGMA). Briefly PLA's principle is based on detecting proteins in close proximity (30–40 nm) by using different species of primary antibodies. A pair of oligonucleotide (PLA probes) detect the primary antibodies bound the proteins of interest and generates a signal only when the two PLA probes have bound in close proximity, meaning that the samples are localized in close proximity. The signal from each detected pair of PLA probes is visualized as an individual fluorescent spot. These PLA signals can be quantified (counted) and assigned to a specific subcellular location based on microscopic images. Images were obtained with a Zeiss Axioplan confocal microscope and PLA spots were counted using Image J software, cells were counted as positive when more than 3 PLA spots were detected. When analyzing pol κ/PCNA interaction a pol κ 'home made' monoclonal antibody from mice purified by Biotem (clone n°16A7-3A11) and PCNA rabbit polyclonal (Abcam ref ab18197) were used. For the quantification of number of foci per nuclei the Cell Profiler program was used to analyze more than 1000 nuclei.

## Quantitative real-time PCR

After transfection with the indicated siRNAs, cells were lysed and total RNA was extracted using TRIzol Reagent (Invitrogen). Total RNA (1 µg) was used as a template for cDNA synthesis utilizing an ImProm-II Reverse Transcription System (Promega) and oligo-dT as a primer. Quantitative real-time PCR was performed using the MX3005P quantitative PCR instrument (Stratagene) with Taq DNA poly-merase (Invitrogen) and SyberGreen and ROX as reference dyes (Invitrogen). Samples were

normalized using GAPDH primers. Both target gene and GAPDH amplification reactions approached 100% efficiency as determined by standard curves. Three independent biological samples were analyzed, and one representative set of results is shown. Primer sequences were as follows: USP1 (Forward): 5'- GGACGCGTTGCTTGGAATGT-3' (Reverse) 5'-TGCCCATCTCAGGGTCTTCA-3'. GADPH: (Forward) 5'-AGCCTCCCGCTTCGCTCTCT-3' (Reverse) 5'-GAGCGATGTGGCTCGGCTGG-3' (*Vallerga et al., 2015*).

## MN assay

Transfected cells were replated at low density. 24 hr after replating cytochalasine B (4.5 µg/ml-Sigma) was added to the media and 40 hr later cells were washed 1 min with hypotonic buffer (KCl 0.0075 M) diluted 1/10 from stock solution in PBS 1X, twice with PBS 1X and fixed with PFA/sucrose 2% for 20 min. DAPI staining served to visualize cell nuclei. 300 binucleated cells were analyzed.

## FISH analysis

Cells were treated with 0.1 µg/ml colcemid (Gibco) for 3 hr. Cells were trypsinized for 1 min trying to take off only the mitotic cells. After centrifugation cell pellets were resuspended in hypotonic solution (0.075 M KCl) and incubated for 15 min at 37°C. Samples were then fixed in a methanol/acetic acid solution (3:1), washed three times in the fixative solution and dropped on slides to obtain spread chromosomes. Metaphase slides were incubated with Rnase (10 µg/ml in 2× SSC) for 1 hr at 37°C followed by dehydration in successive ethanol baths (70, 85, and 100%). The RP 11 36B6 BAC (corresponding to FRA7H locus) was labeled by nick translation (Abbott) with green dUTP (Vysis Spectrum) and ethanol precipitated with human Cot-1 DNA (Roche, California) overnight at −20°C. Precipitated DNA was incubated for 1 hr at 37°C in hybridization mix. The probe was denatured for 8 min at 70°C and applied to denatured metaphases. Samples were incubated over night at 37°C and, after washing (1X SSC, 5 min at 72°C and SCC 1X 5 min at RT) and mounted in VECTASHIELD Antifade Mounting Medium with DAPI. The images were acquired by using an inverted wide-field Zeiss Axio Observer Z1 microscope fitted with a x63 oil NA 1.4 objective and an Axiocam MRm CDD camera.

## Statistical analysis

Frequency distributions of DNA track length were determined with GraphPad Prism 5 software. In non-Gaussian distributions, Mann–Whitney and Kruskal–Wallis tests were used for statistical analyses when comparing two and more than three variables, respectively. Other statistical analyses were performed with GraphPad in Stat software using the Student's t test and the one-way ANOVA test when applicable.

## Acknowledgements

We would like to thank Professors AR Lehmann (University of Sussex) and Maria Cristina Cardoso (Max Delbrück Center for Molecular Medicine, Berlin, Germany) for the gift of reagents.This work was supported by grants from National Institute of Health (R03 TW008924) and the Agencia Nacional de Promoción Científica y Tecnológica (ANPCyT) to VG and from La Ligue Nationale contre le Cancer (Equipe Labellisée) and the Laboratoire d'Excellence Toulouse Cancer LABEX TOUCAN grant (Integrative analysis of resistance in hematological cancers) to JSH. VG and MGB are researchers from the National Council of Scientific and Technological Research (CONICET). SFM and APB are supported by fellowships from CONICET.SFM was also supported by a short-term fellowship from The Journal of Cell Science to visit the JSH laboratory. Christophe Casaux is listed as an author as he contributed actively to this project and sadly passed away before submitting the manuscript.

## Additional information

### Funding

| Funder | Grant reference number | Author |
|---|---|---|
| National Institutes of Health | R03 TW008924 | Vanesa Gottifredi |

| Agencia Nacional de Promo-ción Científica y Tecnológica | PICT-2012-1371 | Vanesa Gottifredi |
|---|---|---|
| Agencia Nacional de Promo-ción Científica y Tecnológica | PICT-2013-1049 | Vanesa Gottifredi |
| Company of Biologists | Travel Fellowship | Sabrina F Mansilla |
| Laboratoire d'Excellence Tou-louse Cancer LABEX TOUCAN | | Jean-Sébastien Hoffmann |
| La Ligue Nationale contra le Cancer | | Jean-Sébastien Hoffmann |

The funders had no role in study design, data collection and interpretation, or the decision to submit the work for publication.

## Author contributions

SFM, APB, Approval of the final version to be published, Conception and design, Acquisition of data, Analysis and interpretation of data, Drafting or revising the article; VB, M-JP, Approval of the final version to be published, Acquisition of data, Analysis and interpretation of data, Drafting or revising the article; MAGB, Approval of the final version to be published, Drafting or revising the article, Contributed unpublished essential data or reagents; CL, SGM, Critical revision of the final draft for important intellectual content, Approval of the final version to be published, Drafting or revising the article, Contributed unpublished essential data or reagents; CC, Conception and design, Contributed unpublished essential data or reagents; J-SH, Approval of the final version to be published, Conception and design, Drafting or revising the article; VG, Approval of the final version to be published, Conception and design, Analysis and interpretation of data, Drafting or revising the article

## Author ORCIDs

Santiago G Miriuka, http://orcid.org/0000-0003-2402-3920

Vanesa Gottifredi, http://orcid.org/0000-0001-9656-5951

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
