## [Decision Letter]

Thank you for submitting your article "Cyclin kinase-independent role of p21CDKN1A in the promotion of nascent DNA elongation in unstressed cells" for consideration by *eLife*. Your article has been favorably evaluated by Tony Hunter as the Senior Editor and three reviewers, one of whom is a member of our Board of Reviewing Editors. The reviewers have opted to remain anonymous.

The reviewers have discussed the reviews with one another and the Reviewing Editor has drafted this decision to help you prepare a revised submission.

The reviewers all found the hypothesis that p21 has a basal role in unstressed cells at the replication fork to keep the Alt polymerases from functioning by direct interactions to be important, and the data showing that such a process is working in cells established for growth in cell culture to be compelling. However, for publication it would be essential to show:

1) That this role is also key in primary cells, and

2) To test the hypothesis further it is key to show that a PIP domain mutation in p21 (ablating interaction sites) cannot complement a null mutation of the endogenous p21 alleles. This experiment could be done in either human or mouse cells.

The reviewers’ comments below are given to provide you with justifications for these issues and have other points that you may wish to address for improving the manuscript in a revision.

*Reviewer #1:*

I found the paper interesting and certainly of potential high impact. So I would like to recommend publication in *eLife* but feel that there are still some additional experiments that need to be done that will make the story more compelling. The key new concept here is that p21 has a basal function in non-stressed cells by interfering with a trans-lesion DNA polymerase κ. In the main system used, an established cell line, the data for a basal activity seems compelling. However, it is essential to show that similar phenotypes for the PIP mutant of PCNA are detected in primary cells or in any convincing way that would eliminate the suggestion that this role for p21 in fact arises because of the initial stress and adaption to cell culture. Technically the authors should also provide more data to show that their siRNA knock-downs are specific or use better still use a knock out PCNA strain (by CRISPR/CAS) with a replacement of the gene with a mutant p21.

Following up on the possibility of selection or adaption for the phenotype, it might be worthwhile for the authors to suggest or discuss how having an alternative polymerase working excessively at forks (without p21) might lead to more ori firing and shorter replication tracks in the first place. Clearly, and just one proposal would be that the "alt polymerase" leads to mistakes and damage stressing the daughter cells with replication pausing which in turn evolves to the need for more ori firing. But is that consistent with what we know about the effects of Pol Κ at the replisome? Might other polymerases be more effective in producing double strand breaks and stress? Or do the authors have a way to explain how it comes to be that more ori's are activated.

*Reviewer #2:*

Mansilla et al. describe a novel function of p21 in preventing genomic instability associated with DNA replication. The authors show that the WT p21, but not the PIP mutant p21 colocalizes with GFP-PCNA. They show that p21 depletion in U2OS and HCT116 cells results in shorter replication tracks without application of exogenous DNA damage and increased origin firing that presumably compensates for the shorter tracks. Cells with decreased levels of p21 also have increased genome instability. Many of the phenotypes were reversible by complementation with WT but not the PIP mutant of p21 or by concomitant depletion of Pol κ. This led the authors to propose that the low levels of p21 present in cells without any exogenous DNA damage and p53 induction are there to protect the PCNA from engaging Pol κ which would result in less processive replication and induction of genomic instability.

This is potentially an interesting set of observations. However, the study is largely done with one cell line, using one siRNA against p21 and Pol κ. Although complementation was performed with p21 cDNA, the key observations should be repeated in p21-/- MEFs or human primary p21-null cells. The most interesting aspect of the study is the rescue of the phenotypes of p21 depletion by Pol κ depletion, but in order for these to be convincing, additional data would be needed. The Pol κ depletion experiments were not done in HCT116 cells and even the efficiency of depletion of Pol Κ was shown only on the exogenously expressed GFP-tagged Pol κ. At least RT-PCR data for Pol κ should be included. In addition, it would be important to show that the levels of Pol κ are increased at the fork in cells lacking p21. This can be done using iPOND with and without p21 depletion.

*Reviewer #3:*

This manuscript presents evidence that the interaction of p21 with PCNA recruits p21 to replication factories, that in the absence of external DNA damage, deletion of p21 impairs normal replication and increases replication stress, that p21 prevents Pol κ (an "alt" or "TLS: polymerase) from contributing to the replication of undamaged DNA, and that these effects depend on the PCNA interacting region domain of p21, but not its CDK-binding domain. These observations are consistent with the authors' hypothesis that p21 has a role in normal replication via an interaction with PCNA. This implies a novel role for p21. In my opinion, the manuscript is well written and should stimulate new studies, and I believe it is worth publishing.

That said, the results obtained here in U2OS and HCT116 cells lines have not yet been confirmed using primary cells, and the evidence for the role of pol κ as the major DNA polymerase involved does not yet exclude roles for several other "alt" polymerases not examined here. These two limitations should be acknowledged, and the language used to express the authors' main claims could be modified at several places (but three examples: Results: "associated exclusively"; "Pol κ is the trigger" – but what about others not examined here (e.g., ζ, ι, ν, θ); Discussion, "Pol κ is the most processive" – not so sure, e.g., what about ζ?)

---

## [Author Response]

*[…] The reviewers all found the hypothesis that p21 has a basal role in unstressed cells at the replication fork to keep the Alt polymerases from functioning by direct interactions to be important, and the data showing that such a process is working in cells established for growth in cell culture to be compelling. However, for publication it would be essential to show:*

*1) That this role is also key in primary cells, and*

*2) To test the hypothesis further it is key to show that a PIP domain mutation in p21 (ablating interaction sites) cannot complement a null mutation of the endogenous p21 alleles. This experiment could be done in either human or mouse cells.*

Primary human cells were used as suggested by the three reviewers and the editor. We used primary human foreskin fibroblast and mesenchymal cells from umbilical cord. As in tumor cells, in such cellular models p21 promotes nascent DNA elongation preventing the accumulation of replication stress and genomic instability markers. This is shown in current Figure 9. To perform such experiments we collaborated with current authors Carlos Luzzani and Santiago Gabriel Miriuka from Laboratorio de Investigaciones Aplicadas en Neurociencias (LIAN-CONICET), FLENI who provided the primary cellular systems and shared their expertise in regards of their manipulation, transfection, etc.

We also provide data demonstrating that only the wt but not the PIP domain mutant complements a null mutation of the endogenous p21 allele. This is showed in Figure 8 panel F to H.

*The reviewers’ comments below are given to provide you with justifications for these issues and have other points that you may wish to address for improving the manuscript in a revision.*

We understand the concerns raised by reviewers and therefore we have addressed each one of their comments. In fact, in order to address reviewers’ comments we have generated 2 Figures (Figure 9 and Figure 6—figure supplement 2) and modified extensively Figure 5 (panels C D and E) and 8 (panels F-H). Detailed responses to each comment are provided below.

*Reviewer #1:*

*I found the paper interesting and certainly of potential high impact. So I would like to recommend publication in eLife but feel that there are still some additional experiments that need to be done that will make the story more compelling. The key new concept here is that p21 has a basal function in non-stressed cells by interfering with a trans-lesion DNA polymerase κ. In the main system used, an established cell line, the data for a basal activity seems compelling. However, it is essential to show that similar phenotypes for the PIP mutant of PCNA are detected in primary cells or in any convincing way that would eliminate the suggestion that this role for p21 in fact arises because of the initial stress and adaption to cell culture.*

As specified above, we have performed the experiments in two primary cells and the data is shown in current Figure 9.

*Technically the authors should also provide more data to show that their siRNA knock-downs are specific or use better still use a knock out PCNA strain (by CRISPR/CAS) with a replacement of the gene with a mutant p21.*

We understand reviewer´s 1 concern. It is true that we have used only one siRNA for p21. However, we also obtained similar results in HCT116 cells in which p21 has been eliminated by homologous recombination (Figure 3—figure supplement 1, Figure 6—figure supplement 2). Moreover, and thanks to the concerted reviewers´ request #2 (mentioned by the editor above), we also included experiments in which an infection with p21wt, but not a PIP defective p21, complements the replication defect of HCT116 p21-/- (see Figure 8). Hence, the specificity of the single p21 siRNA used is supported also by independent lines of evidences in the HCT116 cells.

We however believe that reviewer´s 1 concern should be addressed also for the pol κ siRNA. In this case we used a second siRNA for pol κ and observed similar results to those obtained with the first siRNA specific for pol κ (Figure 6—figure supplement 2). We also used the siRNA for pol κ in HCT116 cells (Figure 6).

*Following up on the possibility of selection or adaption for the phenotype, it might be worthwhile for the authors to suggest or discuss how having an alternative polymerase working excessively at forks (without p21) might lead to more ori firing and shorter replication tracks in the first place. Clearly, and just one proposal would be that the "alt polymerase" leads to mistakes and damage stressing the daughter cells with replication pausing which in turn evolves to the need for more ori firing. But is that consistent with what we know about the effects of Pol Κ at the replisome? Might other polymerases be more effective in producing double strand breaks and stress? Or do the authors have a way to explain how it comes to be that more ori's are activated.*

While the tight association between increased origin firing and reduction in fiber length is well documented in many reports, it is yet unclear which of these two events precede and triggers the other replication defect (e.g. Anglana et al., & Debatisse, Mol Cell 2003 and Pillaire et al. & Hoffmann, Cell Cyle 2007). For example, in the case of Chk1 depletion, the mechanism may involve an initial dysregulation of origin firing which causes a depletion of dNTPs available for nascent DNA elongation (Petterman et al. & Helleday, PNAS, 2010). In contrast, we postulate that the trigger for replication defects in cells depleted from p21 is not origin firing. We propose so because pol κ depletion fully complements the replication defects caused by p21 depletion. Given that the function of pol κ is tightly associated with the elongation of nascent DNA (and not with origin firing), we believe that in the case of this report it is more likely that the primary defect in p21 depleted samples is the slow elongation of nascent DNA at replication forks which then triggers increased origin firing as an attempt to finalize DNA replication of DNA regions which are being duplicated with a slow rate. In fact, we also speculate that such increased origin firing may prevent genomic instability to a certain extent. While the number of slow elongating fibers is frequent (they would not be detected by the stretching assay otherwise), the accumulation of genomic instability events is much less frequent (fold increase in micronuclei 2.1 in p21 depleted cells with respect to p21 positive cells – see Figure 3) and more evident in regions poor in origins as common fragile sites (fold increase of 3.13-see Figure 3). Hence, we believe that in the particular scenario of p21 depletion, the increased origin firing may be compensating replication defects caused by slow nascent DNA elongation.

We also postulate that the shorter fibers are the consequence of a slower DNA synthesis by pol κ, which may be much less processive when compared to replicative polymerases. Our data does not support, as stated in the Discussion, that p21 downregulation causes accumulation of DNA damage. First, in p21 depleted samples, nascent DNA elongation is very steady (but slow) supporting a global change in DNA replication speed rather that the random collision of replication forks with DNA damage. Second, we did not detect phospho-Chk1 accumulation (not shown in the manuscript), (Figure 11) which indicates no signs of global activation of DNA damage response. Last, and more important, the replication defect caused by p21 depletion is eliminated by pol κ depletion. If the defect was caused by DNA lesions, the elimination of an alt polymerase should: a) aggravate the defect (if the alt pol is essential to promote DNA replication across the replication barrier) or b) not alter such defect (if the replication across such barrier could be achieved by another alt polymerase). Hence, our data does not support a model in which p21 depletion generates substrates (DNA lesions) for alt pols. On the contrary, it suggests that undamaged DNA is being misused by the alternative polymerase pol κ as a replication template. This discussion has been incorporated in the manuscript (subsection “The biological consequences of excessive genomic DNA synthesis by Pol κ”).

Author response image 1.U2OS cells were transfected with the indicated siRNAs and 48hs.Later, a western blot was performed to analyse DNA damage markers such as p-Chk1, and p-Kap1. Cells irradiated with 20J/m^2^ were used as a positive control of the DNA damage response.**DOI:**
http://dx.doi.org/10.7554/eLife.18020.021

*Reviewer #2:*

*Mansilla et al. describe a novel function of p21 in preventing genomic instability associated with DNA replication. The authors show that the WT p21, but not the PIP mutant p21 colocalizes with GFP-PCNA. They show that p21 depletion in U2OS and HCT116 cells results in shorter replication tracks without application of exogenous DNA damage and increased origin firing that presumably compensates for the shorter tracks. Cells with decreased levels of p21 also have increased genome instability. Many of the phenotypes were reversible by complementation with WT but not the PIP mutant of p21 or by concomitant depletion of Pol κ. This led the authors to propose that the low levels of p21 present in cells without any exogenous DNA damage and p53 induction are there to protect the PCNA from engaging Pol κ which would result in less processive replication and induction of genomic instability.*

This is potentially an interesting set of observations. However, the study is largely done with one cell line, using one siRNA against p21 and Pol κ. Although complementation was performed with p21 cDNA, the key observations should be repeated in p21-/- MEFs or human primary p21-null cells.

Reviewer 2 is right and his/her request is in agreement with the rest of the reviewers and the editor. As discussed above these experiments were performed in two primary human cells and are summarized in Figure 9.

*The most interesting aspect of the study is the rescue of the phenotypes of p21 depletion by Pol κ depletion, but in order for these to be convincing, additional data would be needed. The Pol κ depletion experiments were not done in HCT116 cells and even the efficiency of depletion of Pol Κ was shown only on the exogenously expressed GFP-tagged Pol κ. At least RT-PCR data for Pol κ should be included.*

In the current version of our manuscript we have included an RT-PCR analysis (see Figure 6—figure supplement 2). Also, we included the requested data in HCT116 cells (see Figure 6—figure supplement 2).

*In addition, it would be important to show that the levels of Pol κ are increased at the fork in cells lacking p21. This can be done using iPOND with and without p21 depletion.*

We believe that this point of reviewer 2 is important. Hence we used two experimental approaches to answer this question. First, we evaluated the interaction of PCNA and GFP-pol κ in the chromatin fraction and found increased interaction of pol κ and PCNA in the absence of p21 (Figure 5). Second, we evaluated the interaction of endogenous pol κ and PCNA in proximity ligation assays finding an increase in the number of PCNA/pol κ interacting foci in the absence of p21 (Figure 5). Such results are also supported by the increased focal organization of GFP-pol κ reported in the original version of the manuscript (current Figure 5). Together, these results support that, in cells with reduced levels of p21, there is an increase in the interaction of PCNA and pol κ in localizations which were previously associated with replication factories.

*Reviewer #3:*

*This manuscript presents evidence that the interaction of p21 with PCNA recruits p21 to replication factories, that in the absence of external DNA damage, deletion of p21 impairs normal replication and increases replication stress, that p21 prevents Pol κ (an "alt" or "TLS: polymerase) from contributing to the replication of undamaged DNA, and that these effects depend on the PCNA interacting region domain of p21, but not its CDK-binding domain. These observations are consistent with the authors' hypothesis that p21 has a role in normal replication via an interaction with PCNA. This implies a novel role for p21. In my opinion, the manuscript is well written and should stimulate new studies, and I believe it is worth publishing.*

That said, the results obtained here in U2OS and HCT116 cells lines have not yet been confirmed using primary cells.

As discussed before this particular point has been raised by the three reviewers and has been addressed using two primary cell lines: human foreskin fibroblast and mesenchymal cells from the umbilical cord (see Figure 9).

And the evidence for the role of pol κ as the major DNA polymerase involved does not yet exclude roles for several other "alt" polymerases not examined here. These two limitations should be acknowledged.

Our point was that the replication stress and genomic instability outputs evaluated in this manuscript were all reverted (surprisingly) by the depletion of one single DNA polymerase, pol κ. We believe that it is possible that other replication associated defects caused by p21 depletion (not explored in this study) could depend on another alternative polymerase. In fact, when using methanol/acetone extraction to reveal alterative pols associated with replication factories (Kannouche et al., Genes & Dev. 2001), we detected increased focal organization of pol κ, η, ι but not in Rev1 (see below). The difference between Rev1 and the other pols may depend on the lack of PIP box in Rev1 (Waters et al& Walker G. Microbiol Mol Biol Rev, 2009). Notably, pol κ is the alt pol with the lowest recruitment to replication factories in control cells (grey bars), and the one with highest fold induction after p21 depletion (Figure 12). In any case as suggested by reviewer 3, it is possible that the dysregulated use of pol η, pol ι and others specialized polymerases could also contribute to the replication defects caused by p21 depletion. Our conclusions are limited to the phenotypes (DNA replication, DNA stress markers, micronuclei and CFS instability) used in this study. Such limitations were carefully stated in the Discussion of our manuscript (see subsection “The complex regulation of Pol κ by p21”).

Author response image 2.(**A**) U2OS cells were transfected with the indicated GFP-pols in siLuc and sip21 depleted cells. Samples were fixed with methanol/acetone and analyzed for focal organization of GFP-pols at replication factories. Representative images for each polymerase are shown. (**B**) Transfected cells were fixed with Methanol/Acetone and the number of cells with detectable foci was quantified.**DOI:**
http://dx.doi.org/10.7554/eLife.18020.022

*And the language used to express the authors' main claims could be modified at several places (but three examples: Results: "associated exclusively"; "Pol κ is the trigger" – but what about others not examined here (e.g., ζ, ι, ν, θ); Discussion, "Pol κ is the most processive" – not so sure, e.g., what about ζ?)*

We have modified the above mentioned sentences according to reviewer´s 3 request (see changes in the subsection “p21 prevents the aberrant use of the alternative DNA polymerase κ during the replication of undamaged DNA”, last paragraph; subsection “The PCNA-binding domain of p21 is necessary and sufficient to prevent the replication defects introduced by Pol κ” and subsection “The complex regulation of Pol κ by p21”.